# Temporal Test-Time Adaptation with State-Space Models

## Abstract

Distribution shifts between training and test data are inevitable over the lifecycle of a deployed model, leading to performance decay. Adapting a model on test samples can help mitigate this drop in performance. However, most test-time adaptation methods have focused on synthetic corruption shifts, leaving a variety of distribution shifts underexplored. In this paper, we focus on distribution shifts that evolve gradually over time, which are common in the wild but challenging for existing methods, as we show. To address this, we propose STAD, a probabilistic state-space model that adapts a deployed model to temporal distribution shifts by learning the time-varying dynamics in the last set of hidden features. Without requiring labels, our model infers time-evolving class prototypes that act as a dynamic classification head. Through experiments on real-world temporal distribution shifts, we show that our method excels in handling small batch sizes and label shift.

## 1 Introduction

Predictive models often have an 'expiration date.' Real-world applications tend to exhibit distribution shift, meaning that the data points seen at test time are drawn from a distribution that is different than the training data's. Moreover, the test distribution usually becomes more unlike the training distribution as time goes on. An example of this is with recommendation systems: trends change, new products are released, old products are discontinued, etc. Unless a model is updated, its ability to make accurate predictions will expire, requiring the model to be taken offline and re-trained. Every iteration of this model life-cycle can be expensive and time consuming. Allowing models to remain 'fresh' for as long as possible is thus an open and consequential problem.

*Test-time adaptation* (TTA) (Liang et al., 2024; Yu et al., 2023) has emerged as a powerful paradigm to preserve model performance under a shifting test distribution. TTA performs online adaptation of a model's parameters using only test-time batches of features. By requiring neither access to labels nor source data, TTA algorithms can be employed in resource-constrained environments, whereas related approaches such as domain generalization, domain adaptation and test-time training cannot. Most TTA methods operate by minimizing an entropy objective (Wang et al., 2021) or updating normalization parameters (Schneider et al., 2020; Nado et al., 2020; Niu et al., 2023).

Synthetically corrupted images (e.g. CIFAR-10-C) are by far the most commonly used benchmark for assessing progress on TTA—with previous work noting a lack of benchmark diversity (Zhao et al., 2023b). These shifts increase the degree of information loss over time, and well-performing TTA methods must learn to preserve a static underlying signal. In this work we focus on a distribution shift of quite a different nature: *Temporal distribution shifts* are due to factors endogenous to the environment and often encode structural change, not just information loss. For example, we will experiment with the functional map of the world (FMoW) benchmark, which has the goal of classifying how land is used as it is developed over time (e.g. rural to urban). Gradual structural change over time has been studied in related subfields, such as temporal domain generalization (Bai et al., 2023), yet received little attention in TTA. We show that this setting, *temporal test-time adaptation* (TempTTA), poses significant challenges for existing TTA methods.

To address this gap, we propose *State-space Test-time Adaptation (STAD)*, a method that builds on the power of probabilistic state-space models (SSMs) to represent non-stationary data distributions over time. STAD dynamically adapts a model's final layer to accommodate an evolving test distribution. Specifically, we employ a SSM to track the evolution of the weight vectors in the final layer, where

each vector represents a class, as distribution shift occurs. For generating predictions on newly acquired test batches, we use the SSM's updated cluster means as the new parameters. STAD leverages Bayesian updating and does not rely upon normalization mechanisms. As a consequence, STAD excels in scenarios where many TTA methods collapse (Niu et al., 2023), such as adapting with very few samples and handling online class imbalance. Our contributions are the following:

- In Sec. 2, we detail the setting of *temporal test-time adaptation* (TempTTA), which aims to cope with shifts that gradually evolve due to variation in the application domain. Despite being ubiquitous in real-world scenarios, these shifts are understudied in the TTA literature and pose significant challenges to established methods, as we demonstrate in Sec. 5.1 (Tab. 2).
- In Sec. 3, we propose STAD, a novel method for TempTTA. It adapts to temporal distribution shifts by modeling its dynamics in representation space. No previous work has explicitly modeled these dynamics, which we demonstrate is crucial via an ablation study (Tab. 5).
- In Sec. 5, we conduct a comprehensive evaluation of STAD and prominent TTA baselines under authentic temporal shifts. Our results show that STAD provides persistent performance gains, particularly in the cases of label shift and small batch sizes.

## 2 PROBLEM SETTING

**Data & Model** We focus on the traditional setting of multi-class classification, where $\mathcal{X} \subseteq \mathbb{R}^D$ denotes the input (feature) space and $\mathcal{Y} \subseteq \{1, \ldots, K\}$ denotes the label space. Let $\mathbf{x}$ and $\mathbf{y}$ be random variables and $\mathbb{P}(\mathbf{x}, \mathbf{y}) = \mathbb{P}(\mathbf{x})\,\mathbb{P}(\mathbf{y}|\mathbf{x})$ the unknown source data distribution. We assume $\boldsymbol{x} \in \mathcal{X}$ and $y \in \mathcal{Y}$ are realisations of $\mathbf{x}$ and $\mathbf{y}$. The goal of classification is to find a mapping $f_\theta$, with parameters $\theta$, from the input space to the label space $f_\theta : \mathcal{X} \to \mathcal{Y}$. Fitting the classifier $f_\theta$ is usually accomplished by minimizing an appropriate loss function (e.g. log loss). Yet, our method is agnostic to how $f_\theta$ is trained and therefore easy to use with, for instance, a pre-trained model from the web.

**Temporal Test-Time Adaptation (TempTTA)** We are interested in adapting a model at test-time to a test distribution that evolves with time. Such temporal distribution shifts have been the focus of work in *temporal domain generalization* (TDG) (Nasery et al., 2021; Qin et al., 2022; Bai et al., 2023; Zeng et al., 2024) and *gradual domain adaptation* (GDA) (Abnar et al., 2021). Our setting differs from TDG and GDA in that we do not require access to the source data and do not alternate the training procedure. While previous work on *continual test-time adaptation* (CTTA) study adaptation to changing target domains, they mostly focus on discrete domains (e.g. different corruption types). We consider a special case of CTTA that is defined by two key aspects: the domain index is temporal and shifts occur gradually over time. See Tab. 1 for a comparison. More formally, let $\mathcal{T} = \{1, \ldots, T\}$ be a set of $T$ time indices. At test time, let the data at time $t \in \mathcal{T}$ be sampled from a distribution $\mathbb{Q}_t(\mathbf{x}, \mathbf{y}) = \mathbb{Q}_t(\mathbf{x})\,\mathbb{Q}_t(\mathbf{y}|\mathbf{x})$. The test distributions differ from the source distribution, $\mathbb{Q}_t(\mathbf{x}, \mathbf{y}) \neq \mathbb{P}(\mathbf{x}, \mathbf{y}) \,\forall t > 0$, and are non-stationary, meaning $\mathbb{Q}_t(\mathbf{x}, \mathbf{y}) \neq \mathbb{Q}_{t'}(\mathbf{x}, \mathbf{y})$ for $t \neq t'$. However, the test distribution evolves gradually such that the discrepancy between consecutive distributions is bounded, $0 \leq d(\mathbb{Q}_t, \mathbb{Q}_{t+1}) \leq \epsilon$, where $d$ is a divergence function (Qin et al., 2022). Like in standard TTA, we of course do not observe labels at test time, and hence we observe only a batch of features $\mathbf{X}_t = \{\mathbf{x}_{1,t}, \ldots, \mathbf{x}_{N,t}\}$, where $\mathbf{x}_{n,t} \sim \mathbb{Q}_t(\mathbf{x})$ (i.i.d.). Given the $t$-th batch of features $\mathbf{X}_t$, the goal is to adapt $f_\theta$, forming a new set of parameters $\theta_t$ such that $f_{\theta_t}$ has better predictive performance on $\mathbf{X}_t$ than $f_\theta$ would have. Since we can only observe features, we assume that the distribution shift must at least take the form of *covariate shift*: $\mathbb{Q}_t(\mathbf{x}) \neq \mathbb{P}(\mathbf{x}) \,\forall t > 0$. In addition, a *label shift* may occur, which poses an additional challenge: $\mathbb{Q}_t(\mathbf{y}) \neq \mathbb{P}(\mathbf{y}) \,\forall t > 0$.

Table 1: TempTTA compared

| | unlabeled test data | source data free | evolving test dist. | time index | bounded shift |
|---|---|---|---|---|---|
| TDG | ✗ | ✗ | ✓ | ✓ | ✓ |
| GDA | ✓ | ✗ | ✗ | ✓ | ✓ |
| TTA | ✓ | ✓ | ✗ | ✗ | ✗ |
| CTTA | ✓ | ✓ | ✓ | ✗ | ✗ |
| **TempTTA** | ✓ | ✓ | ✓ | ✓ | ✓ |

## 3 TRACKING THE DYNAMICS OF TEMPORAL DISTRIBUTION SHIFTS

We now present our method: the core idea is that adaptation to temporal distribution shifts can be done by tracking its gradual change in the model's representations. We employ linear state-space models (SSMs) to capture how test points evolve and drift. The SSM's cluster representations then serve as an adaptive classification head that evolves with the non-stationarity of the distribution shift.

Fig. 1 illustrates our method. In Sec. 3.2, we first introduce the general model and then, in Sec. 3.3, we propose an efficient implementation that leverages the von-Mises-Fisher distribution to model hyperspherical features.

## 3.1 ADAPTATION IN REPRESENTATION SPACE

Following previous work (Iwasawa & Matsuo, 2021; Boudiaf et al., 2022), we adapt only the last layer of the source model. This lightweight approach is reasonable for Temp-TTA since the distribution shifts only gradually, hence constrained adaptation is needed. From a practical perspective, this circumvents backpropagation through potentially large networks such as foundation models and allows adaptation when only embeddings are provided e.g. by an API. More formally, let the classifier $f_\theta$ be a neural network with $L$ total layers. We will treat the first $L-1$ layers, denoted as $f_\theta^{L-1}$, as a black box that transforms the original feature vector $\mathbf{x}$ into a new (lower-dimensional) representation, which we denote as $\mathbf{h}$. The original classifier then maps these representations to the classes as: $\mathbb{E}[y|\mathbf{h}] = \text{softmax}_y(\mathbf{W}_0\mathbf{h})$, where $\text{softmax}_y(\cdot)$ denotes the dimension of the softmax's output corresponding to the $y$-th label index and $\mathbf{W}_0$ are the last-layer weights. As $\mathbf{W}_0$ will only be valid for representations that are similar to the training data, we will discard these parameters when performing TempTTA, learning new parameters $\mathbf{W}_t$ for the $t$-th time step. These new parameters will be used to generate the adapted predictions through the same link function: $\mathbb{E}[y|\mathbf{h}] = \text{softmax}_y(\mathbf{W}_t\mathbf{h})$. In the setting of TempTTA, we observe a batch of features $\mathbf{X}_t$. Passing them through the model yields corresponding representations $\mathbf{H}_t$, and this will be the 'data' used for the probabilistic model we will describe below. Specifically, we will model how the representations change from $\mathbf{H}_t$ to $\mathbf{H}_{t+1}$ next.

## 3.2 A PROBABILISTIC MODEL OF SHIFT DYNAMICS

We now describe our general method for a time-evolving adaptive classification head. We assume that, while the representations $\mathbf{H}_t$ are changing gradually over time, they are still maintaining some class structure in the form of clusters. Our model will seek to track this structure as it evolves. For the intuition of the approach, see Fig. 1. The blue red, and green clusters represent classes of a classification problem. As the distribution shifts from time step $t = 1$ to $t = 3$, the class clusters shift in representation space. Using latent variables $\mathbf{w}_{t,k}$ for the cluster centers, we will assume each representation is drawn conditioned on $K$ latent vectors: $\boldsymbol{h}_{t,n} \sim p(\mathbf{h}_t|\mathbf{w}_{t,1},\ldots,\mathbf{w}_{t,K})$,

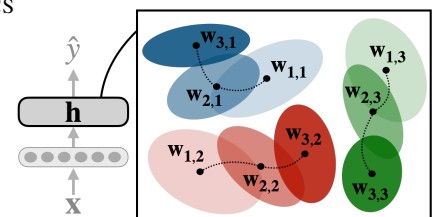

Figure 1: STAD illustrated: It adapts to distribution shifts by inferring dynamic class prototypes $\mathbf{w}_{t,k}$ for each class $k$ (different colors) at each test time point. It operates on the representation space of the penultimate layer.

where $K$ is equal to the number of classes in the prediction task. After fitting the unsupervised model, the $K$ latent vectors will be stacked to create $\mathbf{W}_t$, the last-layer weights of the adapted predictive model (as introduced in Sec. 3.1). We now move on to a technical description.

**Notation and Variables** Let $\mathbf{H}_t = (\mathbf{h}_{t,1},\ldots,\mathbf{h}_{t,N_t}) \in \mathbb{R}^{D \times N_t}$ denote the neural representations for $N_t$ data points at test time $t$. Let $\mathbf{W}_t = (\mathbf{w}_{t,1},\ldots,\mathbf{w}_{t,K}) \in \mathbb{R}^{D \times K}$ denote the $K$ weight vectors at test time $t$. As discussed above, the weight vector $\mathbf{w}_{t,k}$ can be thought of as a latent prototype for class $k$ at time $t$. We denote with $\mathbf{C}_t = (\mathbf{c}_{t,1},\ldots,\mathbf{c}_{t,N_t}) \in \{0,1\}^{K \times N_t}$ the $N_t$ one-hot encoded latent class assignment vectors $\mathbf{c}_{t,n} \in \{0,1\}^K$ at time $t$. The $k$-th position of $\mathbf{c}_{t,n}$ is denoted with $c_{t,n,k}$ and is 1 if $\mathbf{h}_{t,n}$ belongs to class $k$ and 0 otherwise. Like in standard (static) mixture models, the prior of the latent class assignments $p(\mathbf{c}_{t,n})$ is a categorical distribution, $p(\mathbf{c}_{t,n}) = \text{Cat}(\boldsymbol{\pi}_t)$ with $\boldsymbol{\pi}_t = (\pi_{t,1},\ldots,\pi_{t,K}) \in [0,1]^K$ and $\sum_{k=1}^K \pi_{t,k} = 1$. The mixing coefficient $\pi_{t,k}$ gives the a priori probability of class $k$ at time $t$ and can be interpreted as the class proportions. Next, we formally describe how we model the temporal drift of class prototypes.

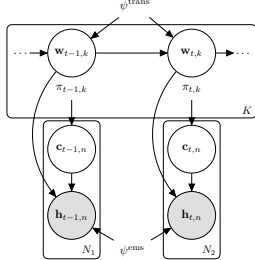

Figure 2: Graphical Model: Representations $\mathbf{h}_{t,n}$ are modeled with a dynamic mixture model. Latent class prototypes $\mathbf{w}_{t,k}$ evolve at each time step, cluster assignments $\mathbf{c}_{t,n}$ determine class membership.

**Dynamics Model** Fig. 2 depicts the plate diagram of our method. We model the evolution of the $K$ prototypes $\mathbf{W}_t = (\mathbf{w}_{t,1}, \ldots, \mathbf{w}_{t,K})$ with $K$ independent Markov processes. The resulting transition model is then:

$$\text{Transition model:} \quad p(\mathbf{W}_t|\mathbf{W}_{t-1}, \psi^{\text{trans}}) = \prod_{k=1}^{K} p(\mathbf{w}_{t,k}|\mathbf{w}_{t-1,k}, \psi^{\text{trans}}), \tag{1}$$

where $\psi^{\text{trans}}$ denote the parameters of the transition density, notably the transition noise. At each time step, the feature vectors $\mathbf{H}_t$ are generated by a mixture distribution over the $K$ classes,

$$\text{Emission model:} \quad p(\mathbf{H}_t|\mathbf{W}_t, \psi^{\text{ems}}) = \prod_{n=1}^{N_t} \sum_{k=1}^{K} \pi_{t,k} \cdot p(\mathbf{h}_{t,n}|\mathbf{w}_{t,k}, \psi^{\text{ems}}). \tag{2}$$

where $\psi^{\text{ems}}$ are the emission parameters. We thus assume at each time step a standard mixture model over the $K$ classes where the class prototype $\mathbf{w}_{t,k}$ defines the latent class center and $\pi_{t,k}$ the mixture weight for class $k$. The joint distribution of representations, prototypes and class assignments can be factorised as follows,

$$p(\mathbf{H}_{1:T}, \mathbf{W}_{1:T}, \mathbf{C}_{1:T}) = p(\mathbf{W}_1) \prod_{t=1}^{T} p(\mathbf{C}_t) p(\mathbf{H}_t|\mathbf{W}_t, \mathbf{C}_t, \psi^{\text{ems}}) \prod_{t=2}^{T} p(\mathbf{W}_t|\mathbf{W}_{t-1}, \psi^{\text{trans}}) \tag{3}$$

$$= \prod_{k}^{K} p(\mathbf{w}_{1,k}) \prod_{t=1}^{T} \prod_{n=1}^{N_t} p(\mathbf{c}_{t,n}) \prod_{k=1}^{K} p(\mathbf{h}_{t,n}|\mathbf{w}_{t,k}, \psi^{\text{ems}})^{\mathbf{c}_{t,n,k}} \prod_{t=2}^{T} \prod_{k=1}^{K} p(\mathbf{w}_{t,k}|\mathbf{w}_{t-1,k}, \psi^{\text{trans}}). \tag{4}$$

We use the notation $\mathbf{H}_{1:T} = \{\mathbf{H}_t\}_{t=1}^{T}$ to denote the representation vectors $\mathbf{H}_t$ for all time steps $T$ and analogously for $\mathbf{W}_{1:T}$ and $\mathbf{C}_{1:T}$. We next outline how we infer the latent class prototypes $\mathbf{W}_{1:T}$.

**Posterior Inference & Adapted Predictions** The primary goal is to update the class prototypes $\mathbf{W}_t$ with the information obtained by the $N_t$ representations of test time $t$. At each test time $t$, we are thus interested in the posterior distribution of the prototypes $p(\mathbf{W}_t|\mathbf{H}_{1:t})$. Once $p(\mathbf{W}_t|\mathbf{H}_{1:t})$ is known, we can update the classification weights with the new posterior mean. The class weights $\mathbf{W}_t$ and class assignments $\mathbf{C}_t$ can be inferred using the Expectation-Maximization (EM) algorithm. In the E-step, we compute $p(\mathbf{W}_{1:T}\mathbf{C}_{1:T}|\mathbf{H}_{1:T})$. In the M-Step, we maximize the expected complete-data log likelihood with respect to the model parameters:

$$\phi^* = \arg\max_{\phi} \mathbb{E}_{p(\mathbf{W}, \mathbf{C}|\mathbf{H})}\big[\log p(\mathbf{H}_{1:T}, \mathbf{W}_{1:T}, \mathbf{C}_{1:T})\big], \tag{5}$$

where $\phi$ denotes the parameters of the transition and emission density as well as the mixing coefficients, $\phi = \{\psi^{\text{trans}}, \psi^{\text{ems}}, \boldsymbol{\pi}_{1:T}\}$. After one optimization step, we collect the $K$ class prototypes into a matrix $\mathbf{W}_t$. Using the same hidden representations used to fit $\mathbf{W}_t$, we generate the predictions using the original predictive model's softmax parameterization:

$$y_{t,n} \sim \texttt{Cat}\big(\mathbf{y}_{t,n}; \texttt{softmax}(\mathbf{W}_t \mathbf{h}_{t,n})\big) \tag{6}$$

where $y_{t,n}$ denotes a prediction sampled for the representation vector $\mathbf{h}_{t,n}$. Note that adaptation can be performed online by optimizing Eqn. (5) incrementally, considering only up to point $t$. To omit computing the complete-data log likelihood for an increasing sequence as time goes on, we employ a sliding window approach.

**Gaussian Model** The simplest parametric form for the transition and emissions models is Gaussian. The transition noise follows a multivariate Gaussian distribution with zero mean and global covariance $\boldsymbol{\Sigma}^{\text{trans}} \in \mathbb{R}^{D \times D}$. The resulting model can be seen as a mixture of $K$ Kalman filters (KFs). For posterior inference, thanks to the linearity and Gaussian assumptions, the posterior expectation $\mathbb{E}_{p(\mathbf{W}, \mathbf{C}|\mathbf{H})}[\cdot]$ in Eqn. (5) can be computed analytically using the well known KF predict, update and smoothing equations (Calabrese & Paninski, 2011; Bishop & Nasrabadi, 2006). However, the closed-form computations come at a cost as they involve inverting matrices of dimensionality $D \times D$. Moreover, the parameter size scales as $K \times D^2$, risking overfitting and consuming substantial memory. These are limitations of the Gaussian formulation making it costly for high-dimensional feature spaces and impractical in low resource environments requiring instantaneous predictions. In the next section, we discuss a model for spherical features that circumvents these limitations.

### 3.3 VON MISES-FISHER MODEL FOR HYPERSPHERICAL FEATURES

Choosing Gaussian densities for the transition and emission models, as discussed above, assumes the representation space follows an Euclidean geometry. However, prior work has shown that assuming the hidden representations lie on the unit *hypersphere* results in a better inductive bias for OOD generalization (Mettes et al., 2019; Bai et al., 2024). This is due to the norms of the representations being biased by in-domain information such as class balance, making angular distances a more reliable signal of class membership in the presence of distribution shift (Mettes et al., 2019; Bai et al., 2024). We too employ the hyperspherical assumption by normalizing the hidden representations such that $||\mathbf{h}||_2 = 1$ and model them with the *von Mises-Fisher* (vMF) distribution (Mardia & Jupp, 2009),

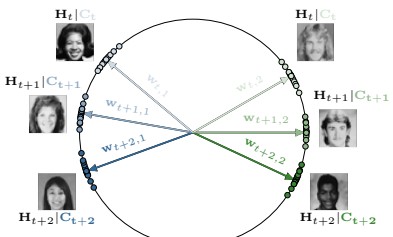

Figure 3: STAD-vMF: Representations lie on the unit sphere. STAD adapts to the distribution shift – induced by changing demographics and styles – by directing the last layer weights $\mathbf{w}_{t,k}$ towards the representations $\mathbf{H}_t$

$$\text{vMF}(\mathbf{h}; \boldsymbol{\mu}_k, \kappa) = C_D(\kappa) \exp\left\{\kappa \cdot \boldsymbol{\mu}_k^T \mathbf{h}\right\} \tag{7}$$

where $\boldsymbol{\mu}_k \in \mathbb{R}^D$ with $||\boldsymbol{\mu}_k||_2 = 1$ denotes the mean direction of class $k$, $\kappa \in \mathbb{R}^+$ the concentration parameter, and $C_D(\kappa)$ the normalization constant. High values of $\kappa$ imply larger concentration around $\boldsymbol{\mu}_k$. The vMF distribution is proportional to a Gaussian distribution with isotropic variance and unit norm. While previous work (Mettes et al., 2019; Ming et al., 2023; Bai et al., 2024) has mainly explored training objectives to encourage latent representations to be vMF-distributed, we apply Eqn. (7) to model the evolving representations.

**Hyperspherical State-Space Model** Returning to the SSM given above, we specify both transition and emission models as vMF distributions, resulting in a hyperspherical transition model, $p(\mathbf{W}_t|\mathbf{W}_{t-1}) = \prod_{k=1}^{K} \text{vMF}(\mathbf{w}_{t,k}|\mathbf{w}_{t-1,k}, \kappa^{\text{trans}})$, and hyperspherical emission model, $p(\mathbf{H}_t|\mathbf{W}_t) = \prod_{n=1}^{N_t} \sum_{k=1}^{K} \pi_{t,k}\text{vMF}(\mathbf{h}_{t,n}|\mathbf{w}_{t,k}, \kappa^{\text{ems}})$. The parameter size of the vMF formulation only scales linearly with the feature dimension, i.e. $\mathcal{O}(DK)$ instead of $\mathcal{O}(D^2K)$ as for the Gaussian case. Notably, the noise parameters, $\kappa^{\text{trans}}, \kappa^{\text{ems}}$ simplify to scalar values which reduces memory substantially. Fig. 3 illustrates this STAD-vMF variant.

**Posterior Inference** Unlike in the linear Gaussian case, the vMF distribution is not closed under marginalization. Consequentially, the posterior distribution required for the expectation in Eqn. (5), $p(\mathbf{W}_{1:T}\mathbf{C}_{1:T}|\mathbf{H}_{1:T})$, cannot be obtained in closed form. We employ a variational EM objective, approximating the posterior with mean-field variational inference, following Gopal & Yang (2014):

$$q(\mathbf{w}_{t,k}) = \text{vMF}(\,\cdot\,; \boldsymbol{\rho}_{t,k}, \gamma_{t,k}) \quad q(\mathbf{c}_{n,t}) = \text{Cat}(\,\cdot\,; \boldsymbol{\lambda}_{n,t}) \quad \forall t, n, k. \tag{8}$$

The variational distribution $q(\mathbf{W}, \mathbf{C})$ factorizes over $n, t, k$ and the objective from Eqn. (5) becomes $\arg\max_\phi \mathbb{E}_{q(\mathbf{W},\mathbf{C})}\left[\log p(\mathbf{H}_{1:T}, \mathbf{W}_{1:T}, \mathbf{C}_{1:T})\right]$. More details including the full maximisation steps for $\phi = \{\kappa^{\text{trans}}, \kappa^{\text{ems}}, \{\{\pi_{t,k}\}_{t=1}^{T}\}_{k=1}^{K}\}$ can be found in App. B.1. Notably, posterior inference for the vMF model is much more scalable than the Gaussian case. It operates with linear complexity in $D$, rather than cubic, reducing runtime significantly. Algorithm 1 (App. B.2) summarizes the method.

**Recovering the Softmax Predictive Distribution** In addition to the inductive bias that is beneficial under distribution shift, using the vMF distribution has an additional desirable property: classification via the cluster assignments is equivalent to the original softmax-parameterized classifier. The equivalence is exact under the assumption of equal class proportions and sharing $\kappa$ across classes:

$$p(\mathbf{c}_{t,n,k} = 1|\mathbf{h}_{t,n}, \mathbf{w}_{t,1}, \ldots, \mathbf{w}_{t,K}, \kappa^{\text{ems}}) = \frac{\text{vMF}(\mathbf{h}_{t,n}; \mathbf{w}_{t,k}, \kappa^{\text{ems}})}{\sum_{j=1}^{K} \text{vMF}(\mathbf{h}_{t,n}; \mathbf{w}_{t,j}, \kappa^{\text{ems}})}$$

$$= \frac{C_D(\kappa^{\text{ems}}) \exp\left\{\kappa^{\text{ems}} \cdot \mathbf{w}_{t,k}^T \mathbf{h}_{t,n}\right\}}{\sum_{j=1}^{K} C_D(\kappa^{\text{ems}}) \exp\left\{\kappa^{\text{ems}} \cdot \mathbf{w}_{t,j}^T \mathbf{h}_{t,n}\right\}} = \text{softmax}\left(\kappa^{\text{ems}} \cdot \mathbf{W}_t^T \mathbf{h}_{t,n}\right), \tag{9}$$

which is equivalent to a softmax with temperature-scaled logits, with the temperature set to $1/\kappa^{\text{ems}}$. Temperature scaling only affects the probabilities, not the modal class prediction. If using class-specific $\kappa^{\text{ems}}$ values and assuming imbalanced classes, then these terms show up as class-specific bias terms: $p(\mathbf{c}_{t,n,k} = 1|\mathbf{h}_{t,n}, \mathbf{w}_{t,1}, \ldots, \mathbf{w}_{t,K}, \kappa_1^{\text{ems}}, \ldots, \kappa_K^{\text{ems}}) \propto \exp\left\{\kappa_k^{\text{ems}} \cdot \mathbf{w}_{t,k}^T \mathbf{h}_{t,n} + \log C_D(\kappa_k^{\text{ems}}) + \log \pi_{t,k}\right\}$ where $C_D(\kappa_k^{\text{ems}})$ is the vMF's normalization constant and $\pi_{t,k}$ is the mixing weight.

## 4 RELATED WORK

**State-Space Models for Deep Learning**   Probabilistic state-space models, and the Kalman filter (Kalman, 1960) in particular, have found diverse use in deep learning as a principled way of updating a latent state with new information. In sequence modelling, filter-based architectures are used to learn the latent state of a trajectory in both discrete (Krishnan et al., 2015; Karl et al., 2017; Fraccaro et al., 2017; Becker et al., 2019) and continuous time (Schirmer et al., 2022; Ansari et al., 2023; Zhu et al., 2023). Recent work on structured state-space models (Gu et al., 2022; Smith et al., 2023; Gu & Dao, 2023) has pushed the state-of-the-art in sequence modeling. However, in this setting, the SSM models the dynamics for an individual sequence while we are interested in modeling the dynamics of the data stream as a whole. The latter is also the subject of supervised online learning, and Chang et al. (2023) and Titsias et al. (2023) employ Kalman filters in the supervised, non-stationary setting. While Titsias et al. (2023) also infer the evolution of a linear classification head with an SSM, they require labels for weight updates, whereas our method is fully unsupervised.

**Domain Generalization (DG)**   The goal of domain generalization (Zhou et al., 2022a) is to learn a predictive model that can generalize well to any unseen domains assuming access to multiple source domains at the training stage. Examples of common approaches are domain-invariant feature learning (Arjovsky et al., 2019) and data augmentation (Zhang, 2018). The most relevant subfield to this work is TDG (Nasery et al., 2021; Qin et al., 2022; Bai et al., 2023; Zeng et al., 2024; Cai et al., 2024), which models dynamics from sequential source domains to generalize to evolving target domains. While both TDG and TempTTA address temporal shifts, TDG operates during training and requires a sequence of labeled source domains, whereas our approach improves the performance of arbitrary pre-trained models at test time using only a stream of unlabeled data.

**Unsupervised Domain Adaptation (UDA)**   Unsupervised Domain adaptation improves generalization by exploiting both labeled source data and unlabeled target data. Most related to our setting is GDA (Hoffman et al., 2014; Wulfmeier et al., 2018; Bobu et al., 2018; Kumar et al., 2020; Abnar et al., 2021; Wang et al., 2022a), which aims to adapt to a target domain by exploiting intermediate domains with bounded distribution shift between source and target. GDA methods rely on access to both source and target data for distribution alignment. However, in many practical scenarios, source data may be unavailable due to e.g. privacy concerns, motivating the need for test-time adaptation.

**Test-Time Adaptation (TTA)**   TTA adapts an off-the-shelf pre-trained model directly during inference without access to source data. Early TTA approaches focused on recalculating batch normalization (BN) statistics from test data (Nado et al., 2020; Schneider et al., 2020). This has often been combined with minimizing entropy (Liang et al., 2020; Wang et al., 2021; Zhang et al., 2022; Yu et al., 2024; Gao et al., 2024). Alternative objectives leverage contrastive learning (Chen et al., 2022), invariance regularization (Nguyen et al., 2023), and Hebbian learning (Tang et al., 2023). While many of these methods can be applied online, they don't fully address the challenges of non-stationary test streams. CTTA aims to handle changing distributions without forgetting source knowledge, using strategies such as episodic resets (Press et al., 2024), student-teacher models (Wang et al., 2022b; Döbler et al., 2023; Brahma & Rai, 2023), masking (Liu et al., 2024), and regularization (Niu et al., 2022; Song et al., 2023). However, temporally correlated data remains challenging as they can cause online class imbalance and disrupt BN statistics, leading to model collapse. Solutions include adapted BN strategies (Zhao et al., 2023a; Lim et al., 2023), reservoir sampling (Gong et al., 2022; Yuan et al., 2023), filtering unreliable samples (Niu et al., 2023), and tracking label distributions (Zhou et al., 2023). Methods that bypass BN and instead adapt the classification head avoid collapse under temporal correlation (Boudiaf et al., 2022; Jang et al., 2023). Most similar to our method, T3A

(Iwasawa & Matsuo, 2021) recomputes prototypes from representations. However, T3A relies upon heuristics while STAD explicitly models dynamics using a SSM.

## 5 EXPERIMENTS

We evaluate our method, STAD, against various baselines on a range of datasets under challenging settings. In Sec. 5.1, we study temporal distribution shifts as defined in Sec. 2, demonstrating the difficulty of the task and STAD's robustness in practical settings. In Sec. 5.2, we go beyond temporal shifts and find that STAD is competitive on reproduction datasets and synthetic corruptions as well. Finally, in Sec. 5.3, we provide insights into STAD's mechanisms, confirming the reliability of its prototypes and highlighting the importance of modeling shift dynamics through an ablation study. We now describe the datasets, source architectures, and baselines. Further details are listed in App. D.

**Datasets**   Our primary focus is on temporal distribution shifts. To further evaluate the effectiveness of our method we also test its performance on reproduction datasets (CIFAR-10-1, ImageNetV2) and standard image corruptions (CIFAR-10-C). Details on those are listed in App. D.1.

- **Yearbook** (Ginosar et al., 2015): a dataset of portraits of American high school students taken across eight decades. Data shift in the students' visual appearance is introduced by changing beauty standards, group norms, and demographic changes. We use the Wild-Time (Yao et al., 2022) pre-processing and evaluation procedure resulting into 33,431 images from 1930 to 2013. Each $32 \times 32$ pixel, grey-scaled image is associated with the student's gender as a binary target label. Images from 1930 to 1969 are used for training; the years 1970 - 2013 for testing.
- **EVIS**: the *evolving image search* (EVIS) dataset (Zhou et al., 2022b) consists of images of 10 electronic product and vehicle categories retrieved from Google search, indexed by upload date. The dataset captures shift caused by rapid technological advancements, leading to evolving designs across time. It includes 57,600 RGB images of 256x256 pixels from 2009 to 2020. Models are trained on images from 2009-2011 and evaluated on images from 2012-2020.
- **FMoW-Time**: the *functional map of the world* (FMoW) dataset (Koh et al., 2021) maps $224 \times 224$ RGB satellite images to one of 62 land-use categories. Distribution shift is introduced by technical advancement and economic growth changing how humans make use of the land. FMoW-Time (Yao et al., 2022) is an adaptation of FMoW-WILDS (Koh et al., 2021; Christie et al., 2018), splitting 141,696 images into a training period (2002-2012) and a testing period (2013-2017).

**Source Architectures and Baselines**   We employ a variety of source architectures to demonstrate the model-agnostic nature of our method. They notably vary in backbone architecture (we use CNN, DenseNet, ResNet, WideResNet) and dimensionality of the representation space (from 32 up to 2048). We list details in App. D.2. In addition to the source model, we compare against 7 baselines representing fundamental approaches to TTA. Five of them adapt the feature extractor: Batch norm adaptation (BN) (Schneider et al., 2020; Nado et al., 2020), TENT (Wang et al., 2021), CoTTA (Wang et al., 2022b), SHOT (Liang et al., 2020) and SAR (Niu et al., 2023). Like our method STAD, two baselines adapt the last linear layer: T3A (Iwasawa & Matsuo, 2021) and LAME (Boudiaf et al., 2022). More details are provided in App. D.3. Batch sizes are the same for all baselines (App. D.4). To ensure optimal performance on newly studied datasets, we conduct an extensive hyperparameter search for each baseline, following the hyperparameters and value ranges suggested in the original papers. We then report the best settings. App. D.4 specifies the grid searches we conducted.

### 5.1 TEMPORAL DISTRIBUTION SHIFTS

We start by evaluating the adaptation abilities to temporal distribution shift on three image classification datasets (Yearbook, EVIS, FMoW-Time), which vary in number of classes (2, 10, 62, respectively), representation dimension (32, 512, 1024, respectively) and shift dynamics (recurring, progressive and rapid, respectively as visible in Fig. 4). For the low-dimensional representations of Yearbook, we also evaluate our computationally costly Gaussian model (STAD-Gauss). We evaluate two settings: **(i) covariate shift with a uniform label distribution** and **(ii) covariate shift with additional shift in the label distribution** $\mathbb{Q}_t(\mathbf{y})$. Having a uniform label distribution—samples are evenly shuffled, making test batches nearly class balanced—has been the standard evaluation setting for TTA. However, particularly in temporal distribution shifts, it is highly unlikely that samples arrive

Table 2: Accuracy on **temporal distribution shifts** and label shifts, averaged over three random seeds. Colors highlight performance that either improves or degrades relative to the source model. Best model in bold, second-best underlined.

| Method | Yearbook covariate shift | Yearbook + label shift | EVIS covariate shift | EVIS + label shift | FMoW-Time covariate shift | FMoW-Time + label shift |
|---|---|---|---|---|---|---|
| Source model | $81.30 \pm 4.18$ | | $56.59 \pm 0.92$ | | $68.94 \pm 0.20$ | |
| *adapt feature extractor* | | | | | | |
| BN | $84.54 \pm 2.10$ | $70.47 \pm 0.33$ | $45.72 \pm 2.79$ | $14.48 \pm 1.02$ | $67.60 \pm 0.44$ | $10.14 \pm 0.04$ |
| TENT | $84.53 \pm 2.11$ | $70.47 \pm 0.33$ | $45.73 \pm 2.78$ | $14.49 \pm 1.02$ | $67.86 \pm 0.54$ | $10.21 \pm 0.01$ |
| CoTTA | $84.35 \pm 2.13$ | $66.12 \pm 0.87$ | $46.13 \pm 2.86$ | $14.71 \pm 1.00$ | $\underline{68.50} \pm 0.25$ | $10.19 \pm 0.04$ |
| SHOT | $85.17 \pm 1.89$ | $70.71 \pm 0.20$ | $45.93 \pm 2.75$ | $14.51 \pm 1.00$ | $68.02 \pm 0.51$ | $10.08 \pm 0.07$ |
| SAR | $84.54 \pm 2.10$ | $70.47 \pm 0.33$ | $45.78 \pm 2.80$ | $14.63 \pm 1.00$ | $67.87 \pm 0.51$ | $10.27 \pm 0.10$ |
| RoTTA | $80.49 \pm 3.48$ | $80.15 \pm 3.50$ | $44.28 \pm 3.02$ | $45.38 \pm 2.88$ | $67.43 \pm 0.67$ | $65.77 \pm 0.68$ |
| *adapt classifier* | | | | | | |
| LAME | $81.60 \pm 3.99$ | $82.70 \pm 4.55$ | $56.67 \pm 0.99$ | $\mathbf{69.37} \pm 5.37$ | $68.32 \pm 0.32$ | $\underline{83.05} \pm 0.48$ |
| T3A | $83.49 \pm 2.55$ | $83.46 \pm 2.59$ | $\mathbf{57.63} \pm 0.77$ | $57.32 \pm 0.77$ | $66.77 \pm 0.26$ | $66.83 \pm 0.27$ |
| **STAD-vMF** (ours) | $\underline{85.50} \pm 1.34$ | $\underline{84.46} \pm 1.19$ | $\underline{56.67} \pm 0.82$ | $\underline{62.08} \pm 1.11$ | $\mathbf{68.87} \pm 0.06$ | $\mathbf{86.25} \pm 1.18$ |
| **STAD-Gauss** (ours) | $\mathbf{86.22} \pm 0.84$ | $\mathbf{84.67} \pm 1.46$ | – | – | – | – |

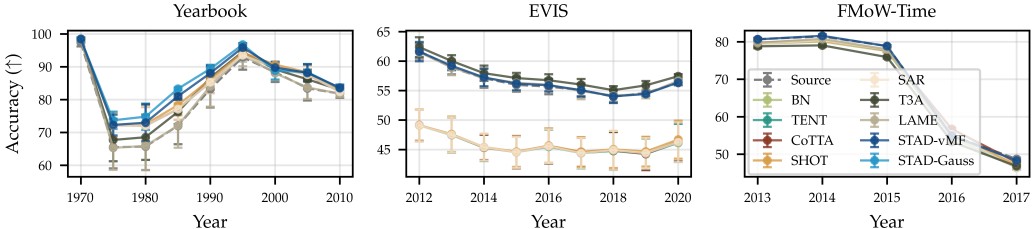

Figure 4: Accuracy over time for temporal distribution shifts: On Yearbook, for instance, STAD mitigates distribution shifts, improving 10 points over the source model for certain years (1980s). Some baselines perform similarly, shown by overlaying accuracy trajectories. Error bars can be tiny.

in this iid-manner. Instead, temporally correlated test streams often observe consecutive samples from the same class (Gong et al., 2022). We follow Lim et al. (2023), ordering the samples by class and thus inducing an extreme label shift. We draw the class order uniformly at random.

**Temporal shifts pose challenges for existing TTA methods.** Tab. 2 shows overall accuracy, averaged over all time steps and three random training seeds. Results that do not outperform the source model are highlighted in red and ones that do in blue. Methods that primarily adapt the feature extractor are shown in the upper section of the table. Ones that, like ours, adapt the classifier are shown in the lower section. To summarize the results: on Yearbook, all methods perform well without label shift, and with label shift, only classifier-based methods improve upon the source baseline. Feature-based methods completely fail on EVIS, and all models, except LAME and STAD-vMF under label shift, fail on FMoW-Time. This leads us to three key takeaways: first, these TempTTA tasks are inherently difficult, leading to smaller adaptation gains overall compared to traditional corruption experiments. Second, methods that adapt only the last layer clearly perform better on temporal distribution shifts under both label distribution settings. This indicates that perhaps 'less is more' for TempTTA. Third, STAD demonstrates the most consistent performance, ranking as the best or second-best model across all datasets and settings. On Yearbook, both the Gaussian and vMF variants outperform the baselines, with the fully parameterized Gaussian model better capturing the distribution shift than the more lightweight vMF model. Fig. 4 displays adaptation performance over different timestamps. We see that on EVIS (*middle*) the methods markedly separate, which reflects the aforementioned gap between feature-based and classifier-based approaches. The reader may also wonder if STAD can be stacked on top of a feature-based approach. We present results exemplary for BN in Tab. 8 (App. E.2) but found that it offers no significant improvement to STAD's performance.

**STAD excels under label shift** Tab. 2 demonstrates that STAD performs particularly well under imbalanced label distributions, delivering the best results on both Yearbook and FMoW-Time. This advantage stems from STAD's clustering approach, where a higher number of samples from the same ground truth class provides a stronger learning signal, leading to more accurate prototype estimates. This is particularly notable on FMoW, where STAD improves upon the source model by more than 17 points. Further, the performance gap between classifier and feature extractor adaptation methods

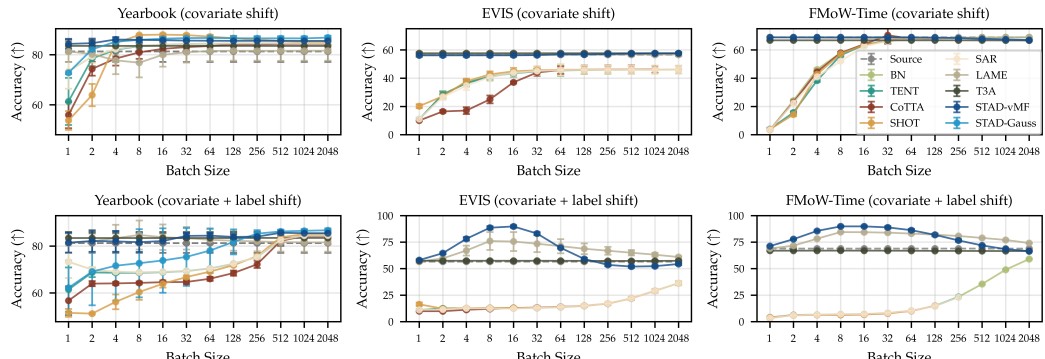

Figure 5: Batch size effects under covariate shift (*first row*) and additional label shift (*second row*): Compared to most baselines, STAD-vMF (dark blue) is relatively robust to very small batch sizes. For label shift on EVIS and FMoW-Time, we observe a sweet spot around batch size 16.

becomes even more pronounced in this setting. This is not surprising as the latter typically depend heavily on current test-batch statistics, making them vulnerable to imbalanced class distributions (Niu et al., 2023). In contrast, having fewer classes to cause confusion allows classifier-based methods to benefit from label shift, with STAD delivering the most persistent adaptation gains.

**STAD is robust to small batch sizes** Adapting to a small number of samples is crucially valuable, as one does not have to wait for a large batch to accumulate in order to make adapted predictions. We next evaluate performance across 12 different batch sizes ranging from 1 to 2048 under both covariate shift and additional label shift. Fig. 5, displays results. STAD-vMF (dark blue line) is able to maintain stable performance under all batch sizes. In Tab. 7 (App. E.1), we report values for batch size 1 showing that STAD adapts successfully even in the most difficult setting. In contrast, methods relying on normalization statistics collapse when not seeing enough samples per adaptation step. For example, on FMoW, feature-based methods collapse to nearly random guessing at the smallest batch sizes. When batch sizes are large, note that TENT, CoTTA, SHOT, and SAR hit memory constraints on FMoW-Time, failing to close the gap to the source model performance in the label shift scenario.

## 5.2 BEYOND TEMPORAL SHIFTS: REPRODUCTION DATASETS AND SYNTHETIC CORRUPTIONS

Although STAD is designed for temporal distribution shifts, we are also interested in the applicability of our method to other types of shifts. Next we report performance on reproduction datasets and synthetic image corruptions.

**Reproduction Datasets** We evaluate our method on reproduction datasets (CIFAR-10.1 and ImageNetV2), which have recently gained attention as benchmarks for more realistic and challenging distribution shifts (Zhao et al., 2023b). We use a batch size of 100 for CIFAR-10.1 and 64 for ImageNetV2. Tab. 3 confirms the difficulty of adapting to more natural distribution shifts. For CIFAR-10.1, only T3A and STAD outperform the source

Table 3: Accuracy on **reproduction datasets** and label shifts, averaged over three random data seeds.

| Method | CIFAR-10.1 covariate shift | CIFAR-10.1 + label shift | ImageNetV2 covariate shift | ImageNetV2 + label shift |
|---|---|---|---|---|
| Source model | 88.25 | | 63.18 | |
| *adapt feature extractor* | | | | |
| BN | $86.45 \pm 0.28$ | $23.83 \pm 0.31$ | $62.69 \pm 0.15$ | $43.20 \pm 0.28$ |
| TENT | $86.75 \pm 0.35$ | $23.87 \pm 0.06$ | $\underline{63.00} \pm 0.16$ | $43.20 \pm 0.28$ |
| CoTTA | $86.75 \pm 0.17$ | $22.37 \pm 0.25$ | $61.66 \pm 0.29$ | $43.73 \pm 0.33$ |
| SHOT | $86.50 \pm 0.23$ | $23.83 \pm 0.31$ | $62.97 \pm 0.22$ | $43.10 \pm 0.34$ |
| SAR | $86.45 \pm 0.28$ | $23.82 \pm 0.33$ | $62.99 \pm 0.10$ | $43.19 \pm 0.25$ |
| RoTTA | $87.17 \pm 0.21$ | $87.85 \pm 0.35$ | $63.39 \pm 0.20$ | $63.20 \pm 0.21$ |
| *adapt classifier* | | | | |
| LAME | $88.20 \pm 0.09$ | $\mathbf{92.42} \pm 0.28$ | $\mathbf{63.15} \pm 0.10$ | $\underline{80.47} \pm 0.32$ |
| T3A | $\underline{88.28} \pm 0.06$ | $89.00 \pm 0.66$ | $62.86 \pm 0.04$ | $63.47 \pm 0.09$ |
| **STAD-vMF** (ours) | $\mathbf{88.42} \pm 0.10$ | $\underline{92.23} \pm 0.70$ | $62.39 \pm 0.05$ | $\mathbf{81.46} \pm 0.24$ |

model for both with and without label shift, with STAD adapting best. For ImageNetV2, none of the methods improve upon the source model when the label distribution is uniform. We again observe that classifier-adaptation methods handle label shifts better by a significant margin.

**Synthetic Corruptions** Lastly, we test our method on gradually increasing noise corruptions of CIFAR-10-C, a standard TTA benchmark. We use a batch size of 100. Tab. 4 shows the accuracy

averaged across all corruption types. We make three key observations. First, performance gains are much higher than on previous datasets indicating the challenge posed by non-synthetic shifts. Second, as expected, methods adapting the backbone model are more performative on input-level noise, since such shifts primarily affect earlier layers. (Tang et al., 2023; Lee et al., 2023). Lastly, STAD is consistently the best method amongst those adapting only the last linear layer.

Table 4: Accuracy on CIFAR-10-C across different levels of **synthetic corruptions**

| Method | Corruption severity | | | | | |
| | 1 | 2 | 3 | 4 | 5 | Mean |
|---|---|---|---|---|---|---|
| Source | 86.90 | 81.34 | 74.92 | 67.64 | 56.48 | 73.46 |
| *adapt feature extractor* | | | | | | |
| BN | 90.18 | 88.16 | 86.24 | 83.18 | 79.27 | 85.41 |
| TENT | **90.87** | **89.70** | 88.32 | 85.89 | 83.09 | 87.57 |
| CoTTA | 90.62 | 89.42 | **88.55** | **87.28** | 85.27 | **88.23** |
| SHOT | 90.31 | 88.66 | 87.31 | 85.02 | 82.13 | 86.69 |
| SAR | 90.16 | 88.09 | 86.26 | 83.32 | 79.48 | 85.46 |
| RoTTA | 90.60 | 89.41 | 88.14 | 85.88 | 83.37 | 87.48 |
| *adapt classifier* | | | | | | |
| LAME | 86.94 | 81.39 | 74.93 | 67.69 | 56.47 | 73.48 |
| T3A | 87.83 | 82.75 | 76.77 | 69.43 | 57.90 | 74.94 |
| **STAD-vMF** (ours) | 88.21 | 83.68 | 78.42 | 72.19 | 62.44 | 76.99 |

## 5.3 ANALYSIS OF TRACKING ABILITIES

Lastly, we seek to further understand the reasons behind STAD's strong performance. At its core, STAD operates through a mechanism of *dynamic clustering*. We next inspect the importance of STAD's dynamics component and assess the fidelity of its clustering.

**Clusters are reliable.** We evaluate how well STAD's inferred cluster centers align with the ground truth cluster centers (computed using labels). We chose the progressively increasing distribution shift of CIFAR-10-C as this dataset represents a bigger challenge for STAD. Fig. 6 *(left)* shows distance (in

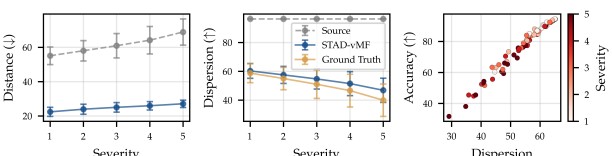

Figure 6: Cluster fidelity on CIFAR-10-C

angular degrees) to the ground truth cluster centers for both the source model and STAD. STAD (blue line) adapts effectively, significantly reducing the angular distance to the ground truth cluster centers. For the source model, the progressive distribution shift causes the ground truth cluster centers to drift increasingly further from the source prototypes (grey line). Additionally, by computing dispersion (Ming et al., 2023) (Fig. 6, *middle*), which measures the spread of the prototypes (in angular degrees), we find that STAD mirrors the ground truth trend (yellow line) of clusters becoming closer together. This is a promising insight, as it suggests that STAD's cluster dispersion could potentially serve as an unsupervised metric to proactively flag when clusters start overlapping and estimate adaptation accuracy. In Fig. 6 *(right)*, we plot accuracy vs dispersion of STAD's prototypes for different corruptions and severity levels, confirming that they positively correlate.

**Dynamics are crucial.** STAD is proposed with the assumption that adapting the class prototypes based on those of the previous time step facilitates rapid and reliable adaptation. However, one could also consider a static version of STAD that does not have a transition model (Eqn. (1)). Rather, the class prototypes are computed as a standard mixture model (Eqn. (2)) and without considering previously inferred pro-

Table 5: Accuracy of dynamic and static versions of STAD (i.e. when removing the transition model)

| Variant | Yearbook | FMoW | CIFAR-10-C |
|---|---|---|---|
| STAD-vMF with dynamics | **85.50** ± 1.30 | **86.25** ± 1.18 | **76.99** |
| STAD-vMF w/o dynamics | 61.03 ± 2.92 | 68.87 ± 0.28 | 73.57 |
| Delta | −24.47 | −17.38 | −3.41 |
| STAD-Gauss with dynamics | **86.22** ± 0.84 | – | – |
| STAD-Gauss w/o dynamics | 57.79 ± 2.14 | – | – |
| Delta | −28.43 | – | – |

totypes. Tab. 5 presents the accuracy differences between the static and dynamic versions of STAD in percentage points. Removing STAD's transition model results in a *substantial performance drop* of up to 28 points. This supports our assumption that SSMs are well-suited for TempTTA.

## 6 CONCLUSION

We have presented *State-space Test-time ADaptation* (STAD), a novel test-time adaptation strategy based on probabilistic state-space models that addresses the challenges of temporal distribution shifts. Our Gaussian and vMF variants of STAD effectively track the evolution of the last layer under distribution shifts, enabling unsupervised adaptation in deployed models. Our extensive experiments highlight the significant challenges that temporal distribution shifts present for existing TTA methods. Future work on TempTTA could explore incorporating temporal information, such as timestamps, to better model the passage of time and extend the approach to irregularly sampled settings.

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

APPENDIX

The appendix is structured as follows:

- App. A provides details on the Gaussian formulation of STAD (STAD-Gauss) stating the employed transition and emission model.
- App. B details the von Mises-Fisher formulation (STAD-vMF). In App. B.1, we provide the inference scheme including all update equations for the variational EM step. In App. B.2, we provide the algorithmic overview of STAD-vMF.
- Sec. 4 discusses further TTA settings that aim for realistic evaluations.
- App. D contains various implementation details.
- App. E shows additional experimental results on single sample adaptation (App. E.1), STAD in combination with BN (App. E.2), domain adaptation benchmarks (App. E.3), comparison to supervised oracle (App. E.4), visualizations of representation space (App. E.5) runtime comparisons (App. E.6) and sensitivity to hyperparameters (App. E.7).

## A   DETAILS ON STAD-GAUSSIAN

### A.1   MODEL FORMULATION

We use a linear Gaussian transition model to describe the weight evolution over time: For each class $k$, the weight vector evolves according to a linear drift parameterized by a class-specific transition matrix $\boldsymbol{A}_k \in \mathbb{R}^{D \times D}$. This allows each class to have independent dynamics. The transition noise follows a multivariate Gaussian distribution with zero mean and global covariance $\boldsymbol{\Sigma}^{\text{trans}} \in \mathbb{R}^{D \times D}$. The transition noise covariance matrix is a shared parameter across classes and time points to prevent overfitting and keep parameter size at bay. Eqn. (10) states the Gaussian transition density.

$$\text{Transition model:} \quad p(\mathbf{W}_t|\mathbf{W}_{t-1}) = \prod_{k=1}^{K} \mathcal{N}(\mathbf{w}_{t,k}|\boldsymbol{A}_k \mathbf{w}_{t-1,k}, \boldsymbol{\Sigma}^{\text{trans}}) \tag{10}$$

$$\text{Emission model:} \quad p(\mathbf{H}_t|\mathbf{W}_t) = \prod_{n=1}^{N_t} \sum_{k=1}^{K} \pi_{t,k} \mathcal{N}(\mathbf{h}_{t,n}|\mathbf{w}_{t,k}, \boldsymbol{\Sigma}^{\text{ems}}) \tag{11}$$

Eqn. (11) gives the emission model of the observed features $\mathbf{H}_t$ at time $t$. As in Eqn. (2), the features at a given time $t$ are generated by a mixture distribution with mixing coefficient $\pi_{t,k}$. The emission density of each of the $K$ component is a multivariate normal with the weight vector of class $k$ at time $t$ as mean and $\boldsymbol{\Sigma}^{\text{ems}} \in \mathbb{R}^{D \times D}$ as class-independent covariance matrix. The resulting model can be seen as a mixture of $K$ Kalman filters. Variants of it has found application in applied statistics (Calabrese & Paninski, 2011).

**Posterior inference**   We use the EM objective of Eqn. (5) to maximize for the model parameters $\phi = \{\{\boldsymbol{A}_k, \{\pi_{t,k}\}_{t=1}^{T}\}_{k=1}^{K}, \boldsymbol{\Sigma}^{\text{trans}}, \boldsymbol{\Sigma}^{\text{ems}}\}$. Thanks to the linearity and Gaussian assumptions, the posterior expectation $\mathbb{E}_{p(\mathbf{W},\mathbf{C}|\mathbf{H})}[\cdot]$ in Eqn. (5) can be computed analytically using the well known Kalman filter predict, update and smoothing equations (Calabrese & Paninski, 2011; Bishop & Nasrabadi, 2006).

**Complexity**   The closed form computations of the posterior $p(\mathbf{W}_t|\mathbf{H}_{1:t})$ and smoothing $p(\mathbf{W}_t|\mathbf{H}_{1:T})$ densities come at a cost as they involve amongst others matrix inversions of dimensionality $D \times D$. This results in considerable computational costs and can lead to numerical instabilities when feature dimension $D$ is large. In addition, the parameter size scales $K \times D^2$ risking overfitting and consuming substantial memory. These are limitations of the Gaussian formulation making it costly for high-dimensional feature spaces and impractical in low resource environments requiring instant predictions.

# B  DETAILS ON STAD-VMF

## B.1  INFERENCE

**Complete-data log likelihood**  Using the von Mises-Fisher distribution as hyperspherical transition and emission model, the log of the complete-data likelihood in Eqn. (3) becomes

$$\log p(\mathbf{H}_{1:T}, \mathbf{W}_{1:T}, \mathbf{C}_{1:T}) = \sum_{k}^{K} \log p(\mathbf{w}_{1,k}) \tag{12}$$

$$+ \sum_{t=1}^{T} \sum_{n=1}^{N_t} \log p(\mathbf{c}_{t,n}) + \sum_{k=1}^{K} \mathbf{c}_{t,n,k} \log p(\mathbf{h}_{t,n} | \mathbf{w}_{t,k}, \kappa^{\mathrm{ems}}) \tag{13}$$

$$+ \sum_{t=2}^{T} \sum_{k=1}^{K} \log p(\mathbf{w}_{t,k} | \mathbf{w}_{t-1,k}, \kappa^{\mathrm{trans}}) \tag{14}$$

$$= \sum_{k}^{K} \log C_D(\kappa_{0,k}) + \kappa_{0,k} \boldsymbol{\mu}_{0,k}^T \mathbf{w}_{1,k} \tag{15}$$

$$+ \sum_{t=1}^{T} \sum_{n=1}^{N_t} \sum_{k=1}^{K} c_{n,t,k} \big( \log \pi_{t,k} + \log C_D(\kappa^{\mathrm{ems}}) + \kappa^{\mathrm{ems}} \mathbf{w}_{t,k}^T \mathbf{h}_{t,n} \big) \tag{16}$$

$$+ \sum_{t=2}^{T} \sum_{k=1}^{K} \log C_D(\kappa^{\mathrm{trans}}) + \kappa^{\mathrm{trans}} \mathbf{w}_{t-1,k}^T \mathbf{w}_{t,k} \tag{17}$$

where $\kappa_{0,k}$ and $\boldsymbol{\mu}_{0,k}$ denote the parameters of the first time step. In practise, we set $\boldsymbol{\mu}_{0,k}$ to the source weights and $\kappa_{0,k} = 100$ (see App. D).

**Variational EM objective**  As described in Sec. 3.3, we approximate the posterior $p(\mathbf{W}_{1:T}, \mathbf{C}_{1:T} | \mathbf{H}_{1:T})$ with a variational distribution $q(\mathbf{W}_{1:T}, \mathbf{C}_{1:T})$ assuming the factorised form

$$q(\mathbf{W}_{1:T}, \mathbf{C}_{1:T}) = \prod_{t=1}^{T} \prod_{k=1}^{K} q(\mathbf{w}_{t,k}) \prod_{n=1}^{N_t} q(\mathbf{c}_{n,t}), \tag{18}$$

where we parameterise $q(\mathbf{w}_{t,k})$ and $q(\mathbf{c}_{n,t})$ with

$$q(\mathbf{w}_{t,k}) = \mathtt{vMF}(\,\cdot\,; \boldsymbol{\rho}_{t,k}, \gamma_{t,k}) \quad q(\mathbf{c}_{n,t}) = \mathtt{Cat}(\,\cdot\,; \boldsymbol{\lambda}_{n,t}) \quad \forall t, n, k. \tag{19}$$

We obtain the variational EM objective

$$\arg\max_{\phi} \mathbb{E}_q \big[ \log p(\mathbf{H}_{1:T}, \mathbf{W}_{1:T}, \mathbf{C}_{1:T}) \big], \tag{20}$$

where $\mathbb{E}_{q(\mathbf{W}_{1:T}, \mathbf{C}_{1:T})}$ is denoted $\mathbb{E}_q$ to reduce clutter.

**E-step**  Taking the expectation of the complete-data log likelihood (Eqn. (12)) with respect to the variational distribution (Eqn. (18)) gives

$$\mathbb{E}_q[\log p(\mathbf{H}_{1:T}, \mathbf{W}_{1:T}, \mathbf{C}_{1:T})] = \sum_{k}^{K} \log C_D(\kappa_{0,k}) + \kappa_{0,k} \boldsymbol{\mu}_{0,k}^T \mathbb{E}_q[\mathbf{w}_{1,k}] \tag{21}$$

$$+ \sum_{t=1}^{T} \sum_{n=1}^{N_t} \sum_{k=1}^{K} \mathbb{E}_q[c_{n,t,k}] \big( \log \pi_{t,k} + \log C_D(\kappa^{\mathrm{ems}}) + \kappa^{\mathrm{ems}} \mathbb{E}_q[\mathbf{w}_{t,k}]^T \mathbf{h}_{t,n} \big) \tag{22}$$

$$+ \sum_{t=2}^{T} \sum_{k=1}^{K} \log C_D(\kappa^{\mathrm{trans}}) + \kappa^{\mathrm{trans}} \mathbb{E}_q[\mathbf{w}_{t-1,k}]^T \mathbb{E}_q[\mathbf{w}_{t,k}] \tag{23}$$

Solving for the variational parameters, we obtain

$$\lambda_{n,t,k} = \frac{\beta_{n,t,k}}{\sum_{j=1}^{K} \beta_{n,t,j}} \quad \text{with} \quad \beta_{n,t,k} = \pi_{t,k} C_D(\kappa^{\text{ems}}) \exp(\kappa^{\text{ems}} \mathbb{E}_q[\mathbf{w}_{t,k}]^T \mathbf{h}_{n,t}) \tag{24}$$

$$\boldsymbol{\rho}_{t,k} = \frac{\kappa^{\text{trans}} \mathbb{E}_q[\mathbf{w}_{t-1,k}] + \kappa^{\text{ems}} \sum_{n=1}^{N_t} \mathbb{E}_q[c_{n,t,k}] \mathbf{h}_{n,t} + \kappa^{\text{trans}} \mathbb{E}_q[\mathbf{w}_{t+1,k}]}{\gamma_{t,k}} \tag{25}$$

$$\gamma_{t,k} = ||\boldsymbol{\rho}_{t,k}|| \tag{26}$$

The expectations are given by

$$\mathbb{E}[c_{n,t,k}] = \lambda_{n,t,k} \tag{27}$$

$$\mathbb{E}[\mathbf{w}_{t,k}] = A_D(\gamma_{t,k}) \boldsymbol{\rho}_{t,k}, \tag{28}$$

where $A_D(\kappa) = \frac{I_{D/2}(\kappa)}{I_{D/2-1}(\kappa)}$ and $I_v(a)$ denotes the modified Bessel function of the first kind with order $v$ and argument $a$.

**M-step** Maximizing objective (Eqn. (20)) with respect to the model parameters $\phi = \{\kappa^{\text{trans}}, \kappa^{\text{ems}}, \{\pi_{t,k}\}_{t=1}^{T}\}_{k=1}^{K}\}$ gives

$$\hat{\kappa}^{\text{trans}} = \frac{\bar{r}^{\text{trans}} D - (\bar{r}^{\text{trans}})^3}{1 - (\bar{r}^{\text{trans}})^2} \quad \text{with} \quad \bar{r}^{\text{trans}} = \left\| \frac{\sum_{t=2}^{T} \sum_{k=1}^{K} \mathbb{E}_q[\mathbf{w}_{t-1,k}]^T \mathbb{E}_q[\mathbf{w}_{t,k}]}{(T-1) \times K} \right\| \tag{29}$$

$$\hat{\kappa}^{\text{ems}} = \frac{\bar{r}^{\text{ems}} D - (\bar{r}^{\text{ems}})^3}{1 - (\bar{r}^{\text{ems}})^2} \quad \text{with} \quad \bar{r}^{\text{ems}} = \left\| \frac{\sum_{t=2}^{T} \sum_{k=1}^{K} \sum_{n=1}^{N_t} \mathbb{E}_q[\mathbf{c}_{n,t,k}] \mathbb{E}_q[\mathbf{w}_{t,k}]^T \mathbf{h}_{n,t}}{\sum_{t=1}^{T} N_t} \right\| \tag{30}$$

$$\pi_{t,k} = \frac{\sum_{n=1}^{N_t} \mathbb{E}[c_{n,t,k}]}{N_t} \tag{31}$$

Here we made use of the approximation from Banerjee et al. (2005) to compute an estimate for $\kappa$,

$$\hat{\kappa} = \frac{\bar{r} D - \bar{r}^3}{1 - \bar{r}^2} \quad \text{with} \quad \bar{r} = A_D(\hat{\kappa}). \tag{32}$$

## B.2 ALGORITHMIC OVERVIEW

---

**Algorithm 1** STAD-vMF

---

1: Input: source model $f_\theta$, test batches $\{\mathbf{X}_t\}_{t=1}^T$, sliding window size $s$
2: Initialize: mixing coefficients $\pi_{t,k} \leftarrow \frac{1}{K} \forall t, k$, weights $\mathbf{W}_t \leftarrow \mathbf{W}_0 \forall t$, transition concentration $\kappa^{trans}$ and emission concentration $\kappa^{ems}$
3: **for** $t \in \mathcal{T}$ **do**
4:     Define sliding window $S_t = \{\tau \mid \max(1, t-s) \leq \tau \leq t\}$
5:     Get representations $\mathbf{H}_t \leftarrow f_\theta^{L-1}(\mathbf{X}_t)$
    E-step:
6:     **for** $\tau \in S_t$ **do**
7:         **for** $k = 1, \ldots, K$ **do**
8:             **for** $n = 1, \ldots, N_t$ **do**
9:                 Compute $\mathbb{E}[c_{n,\tau,k}]$ by Eqn. (27)        ▷ Cluster assignments
10:             **end for**
11:             Compute $\mathbb{E}[\mathbf{w}_{\tau,k}]$ by Eqn. (28)        ▷ Prototype dynamics
12:         **end for**
13:     **end for**
    M-Step:
14:     **for** $\tau \in S_t$ **do**
15:         Compute $\pi_\tau$ by Eqn. (31)        ▷ Label distribution
16:     **end for**
17:     Compute $\kappa^{\mathrm{ems}}$ by Eqn. (30) (optional)
18:     Compute $\kappa^{\mathrm{trans}}$ by Eqn. (29) (optional)
    Predict:
19:     Get predictions $\{y_{t,n}\}_{n=1}^{N_t}$ by Eqn. (6)
20: **end for**
21: **return** Predictions $\{y_{t,n}\}_{n=1}^{N_t}$

---

## C FURTHER RELATED WORK ON REALISTIC TTA

Realistic TTA aims to provide evaluation settings that reflect challenging test conditions possibly encountered in the real world. Past work mostly focuses on the ordering of samples in a test stream taking into account two main components: the domain index and the class index. In the standard setting (fully TTA) the test stream contains a single domain and test data is iid sampled resulting in a uniform label distribution per batch. Continual TTA (Wang et al., 2022b) expands on this setting by considering several domains sequentially. Non-iid TTA (Gong et al., 2022) is another extension of Fully TTA that challenges the iid assumption by introducing temporal correlation in the sampling procedure resulting in class imbalances per test batch. The Practical TTA setting (Yuan et al., 2023) combines CTTA and Non-iid TTA and considers continually changing domains and temporal correlation simultaneously. TRIBE [2] adds on Practical TTA by also controlling global class imbalance over the entire data stream. Instead of regrouping the data stream by domain, ROID (Marsden et al., 2024) introduces mixed domains per batch as an additional option for both iid and non-iid sampled class labels. Lastly UniTTA (Du et al., 2024) comprises a combined set of 36 sampling strategies considering both ordering and imbalance of both domains and class labels.

## D  Experimental Details

We next list details on the experimental setup and hyperparameter configurations. All experiments are performed on NVIDIA RTX 6000 Ada with 48GB memory.

### D.1  Datasets beyond Temporal Distribution Shifts

- **CIFAR-10.1** (Recht et al., 2019): a reproduction of CIFAR-10 (Krizhevsky et al., 2009) assembled from the same data source by following the same cleaning procedure. The dataset contains 2,000 $32 \times 32$ pixel images of 10 classes. Models are trained on the original CIFAR-10 train set.
- **ImageNetV2** (Recht et al., 2019): a reproduction of ImageNet (Deng et al., 2009) with 10,000 images of 1,000 classes scaled to $224 \times 224$ pixels. Models are trained on the original ImageNet.
- **CIFAR-10-C**: a dataset derived from CIFAR-10, to which 15 corruption types are applied with 5 severity levels (Hendrycks & Dietterich, 2019). We mimic a gradual distribution shift by increasing the corruption severity starting from the lowest level (severity 1) to the most sever corruption (severity 5). This results in a test stream of $5 \times 10,000$ images per corruption type.

### D.2  Source Architectures

- **CNN**: We employ the four-block convolutional neural network trained by Yao et al. (2022) to perform the binary gender prediction on the yearbook dataset. Presented results are averages over three different seeds trained with empirical risk minimization. The dimension of the latent representation space is 32.
- **WideResNet**: For the CIFAR-10 experiments, we follow Song et al. (2023); Wang et al. (2021) and use the pre-trained WideResNet-28 (Zagoruyko & Komodakis, 2016) model from the RobustBench benchmark (Croce et al., 2021). The latent representation have a dimension of 512.
- **DenseNet**: For FMoW-Time, we follow the backbone choice of Koh et al. (2021); Yao et al. (2022) and use DenseNet121 (Huang et al., 2017) for the land use classification task. Weights for three random trainings seeds are provided by Yao et al. (2022). We use the checkpoints for plain empirical risk minimization. The latent representation dimension is 1024.
- **ResNet**: For EVIS, we follow Zhou et al. (2022b) and use their ResNet-18 (He et al., 2016) model with a representation dimension of 512 and train on three random seeds. For ImageNet, we follow Song et al. (2023) and employ the standard pre-trained ResNet-50 model from RobustBench (Croce et al., 2021). Latent representations are of dimension 2048.

### D.3  Baselines

- **Source Model**: the un-adapted original model.
- **BatchNorm (BN) Adaptation** (Schneider et al., 2020; Nado et al., 2020): aims to adapt the source model to distributions shift by collecting normalization statistics (mean and variance) of the test data.
- **Test Entropy Minimization (TENT)** (Wang et al., 2021): goes one step further and optimizes the BN transformation parameters (scale and shift) by minimizing entropy on test predictions.
- **Continual Test-Time Adaptation (CoTTA)** (Wang et al., 2022b): takes a different approach by optimizing all model parameters with an entropy objective on augmentation averaged predictions and combines it with stochastic weight restore to prevent catastrophic forgetting.
- **Source HypOthesis Transfer (SHOT)** (Liang et al., 2020) adapts the feature extractor via an information maximization loss in order to align the representations with the source classifier.
- **Sharpness-Aware Reliable Entropy Minimization (SAR)** (Niu et al., 2023) filters out samples with large gradients based on their entropy values and encourages convergence to a flat minimum.
- **Laplacian Adjusted Maximum likelihood Estimation (LAME)** (Boudiaf et al., 2022) regularizes the likelihood of the source model with a Laplacian correction term that encourages neighbouring representations to be assigned to the same class.
- **Test-Time Template Adjuster (T3A)** (Iwasawa & Matsuo, 2021) computes new class prototypes by a running average of low entropy representations.

### D.4 IMPLEMENTATION DETAILS AND HYPERPARAMETERS

By the nature of test-time adaptation, choosing hyperparameters is tricky (Zhao et al., 2023b) since one cannot assume access to a validation set of the test distribution in practise. To ensure we report the optimal performance on new or barely used datasets (Yearbook, EVIS, FMoW, CIFAR-10.1 and ImageNetV2), we perform a grid search over hyperparameters as suggested in the original papers. We perform separate grid searches for the uniform label distribution and online imbalanced label distribution setting. Reported performance correspond to the best setting. If the baselines were studied in the gradual CIFAR-10-C setting by Wang et al. (2022b), we use their hyperparameter setup; otherwise, we conduct a grid search as described earlier. Unless there is a built-in reset (SAR) or convergence criteria (LAME) all methods run without reset and one optimization step is performed. We use the same batch sizes for all baselines. For Yearbook we comprise all samples of a year in one batch resulting in a batch size of 2048. To create online class imbalance, we reduce the batch size to 64. We use a batch size of 100 for EVIS, CIFAR.10.1 and CIFAR-10-C and 64 for FMoW-Time and ImageNetV2.

**BN** (Schneider et al., 2020; Nado et al., 2020) Normalization statistics during test-time adaptation are a running estimates of both the training data and the incoming test statistics. No hyperparameter optimization is necessary here.

**TENT** (Wang et al., 2021) Like in BN, the normalization statistics are based on both training and test set. As in Wang et al. (2021), we use the same optimizer settings for test-time adaptation as used for training, except for the learning rate that we find via grid search on $\{1e^{-3}, 1e^{-4}, 1e^{-5}, 1e^{-6}, 1e^{-7}\}$. Adam optimizer (Kingma & Ba, 2015) is used. For CIFAR-10-C, we follow the hyperparameter setup of Wang et al. (2022b) and use Adam optimizer with learning rate $1e-3$.

**CoTTA** (Wang et al., 2022b) We use the same optimizer as used during training (Adam optimizer Kingma & Ba (2015)). For hyperparameter optimization we follow the parameter suggestions by Wang et al. (2022b) and conduct a grid search for the learning rate ($\{1e^{-3}, 1e^{-4}, 1e^{-5}, 1e^{-6}, 1e^{-7}\}$), EMA factor ($\{0.99, 0.999, 0.9999\}$) and restoration factor ($\{0, 0.001, 0.01, 0.1\}$). Following Wang et al. (2022b), we determine the augmentation confidence threshold by the 5% percentile of the softmax prediction confidence from the source model on the source images. For the well-studied CIFAR-10-C dataset, we follow the setting of Wang et al. (2022b) and use Adam optimizer with learning rate $1e-3$. The EMA factor is set to $0.999$, the restoration factor is $0.01$ and the augmentation confidence threshold is $0.92$.

**SHOT** (Liang et al., 2020) We perform a grid search for the learning rate over $\{1e-3, 1e-4\}$ and for $\beta$, the scaling factor for the loss terms, over $\{0.1, 0.3\}$.

**SAR** (Niu et al., 2023) We conduct a grid search over the learning rate selecting among $\{1e-2, 1e-3, 1e-4, 0.00025\}$. Like the authors, we compute the $E_0$ threshold as a function of number of classes $0.4 \times \ln K$, use SDG, a moving average factor of $0.9$, and the reset threshold of $0.2$. The updated layers include all of batch, group, and layer normalization where present in the source model.

**LAME** (Boudiaf et al., 2022) The only hyperparameter is the choice of affinity matrix. Like Boudiaf et al. (2022) we use a $k$-NN affinity matrix and select the number of nearest neighbours among $\{1, 3, 5\}$.

**T3A** (Iwasawa & Matsuo, 2021) We test different values for the hyperparameter $M$. The $M$-th largest entropy values are included in the support set used for computing new prototypes. We test the values $\{1, 5, 20, 50, 100, \text{None}\}$. None corresponds to no threshold, i.e. all samples are part of the support set.

**STAD-vMF** The hyperparameters are the initialization values of the transition concentration parameter $\kappa^{trans}$, emission concentration parameter $\kappa^{ems}$ and the sliding window size $s$. We chose the concentration parameters from $\{100, 1000\}$. Tab. 6 lists employed settings. We use a default window size of $s = 3$. For Yearbook, we employ class specific noise parameters $\kappa_k^{\text{trans}}$ and $\kappa_k^{\text{ems}}$ as discussed in Sec. 3.3. For the other datasets, we found a more restricted noise model beneficial.

Particularly, we use global concentration parameters, $\kappa^{\mathrm{trans}}$ and $\kappa^{\mathrm{ems}}$, and follow suggestions by Gopal & Yang (2014) to keep noise concentration parameters fixed instead of learning them via maximum likelihood (see line 17 and 18 in Algorithm 1). Keeping them fix acts as a regularization term as it controls the size of the cluster (via $\kappa^{\mathrm{ems}}$) and the movement of the prototypes (via $\kappa^{\mathrm{trans}}$). Low concentration values generally correspond to more adaptation flexibility while larger values results in a more conservative and rigid model.

**STAD-Gauss** We initialize the mixing coefficients with $\pi_{t,k} = \frac{1}{K} \forall t, k$, the transition covariance matrix with $\mathbf{\Sigma}^{\mathrm{trans}} = 0.01 \times \mathbf{I}$ and the emission covariance matrix with $\mathbf{\Sigma}^{\mathrm{ems}} = 0.5 \times \mathbf{I}$. We found a normalization of the representations to be also beneficial for STAD-Gauss. Note that despite normalization, the two models are not equivalent. STAD-Gauss models the correlation between different dimensions of the representations and is therefore more expressive, while STAD-vMF assumes an isotropic variance.

Table 6: Hyperparameters employed for STAD-vMF

| Dataset | $\kappa^{trans}$ | $\kappa^{ems}$ |
|---|---|---|
| Yearbook (covariate shift) | 100 | 100 |
| Yearbook (+ label shift) | 1000 | 100 |
| EVIS (covariate shift) | 1000 | 1000 |
| EVIS (+ label shift) | 1000 | 100 |
| FMoW-Time (covariate shift) | 100 | 100 |
| FMoW-Time (+ label shift) | 1000 | 100 |
| CIFAR-10.1 (covariate shift) | 1000 | 1000 |
| CIFAR-10.1 (+ label shift) | 1000 | 100 |
| ImageNetV2 (covariate shift) | 100 | 1000 |
| ImageNetV2 (+ label shift) | 1000 | 100 |
| CIFAR-10-C (covariate shift) | 1000 | 100 |

# E  ADDITIONAL RESULTS

## E.1  SINGLE SAMPLE ADAPTATION

Table 7: Adaptation accuracy on temporal distribution shift with single sample adaptation (batch size 1) for both covariate shift and additional online label shift: Table shows values as plotted in Fig. 5. Most methods collapse when only provided with one sample per adaptation step. STAD can improve upon the source model in 4 out of 6 scenarios.

| Model | Yearbook covariate shift | + label shift | EVIS covariate shift | + label shift | FMoW covariate shift | + label shift |
|---|---|---|---|---|---|---|
| Source | 81.30 ± 4.18 | | 56.59 ± 0.92 | | 68.94 ± 0.20 | |
| BN | 73.32 ± 6.90 | 73.32 ± 6.90 | 11.12 ± 0.97 | 11.12 ± 0.97 | 3.46 ± 0.03 | 3.46 ± 0.03 |
| TENT | 61.33 ± 9.42 | 61.45 ± 9.46 | 10.80 ± 0.84 | 10.78 ± 0.81 | 4.00 ± 0.87 | 3.99 ± 0.87 |
| CoTTA | 55.91 ± 5.26 | 56.71 ± 6.12 | 10.04 ± 0.08 | 9.88 ± 0.34 | 3.42 ± 0.14 | 3.42 ± 0.03 |
| SHOT | 53.71 ± 3.77 | 51.48 ± 1.68 | 20.23 ± 1.40 | 16.36 ± 1.67 | 3.84 ± 0.21 | 4.01 ± 0.44 |
| SAR | 73.32 ± 6.89 | 73.38 ± 7.01 | 11.12 ± 0.97 | 11.12 ± 0.97 | 3.46 ± 0.03 | 3.46 ± 0.03 |
| T3A | 83.51 ± 2.54 | 83.44 ± 2.60 | 57.63 ± 0.77 | 57.40 ± 0.76 | 66.78 ± 0.24 | 66.87 ± 0.27 |
| LAME | 81.29 ± 4.18 | 81.30 ± 4.18 | 56.59 ± 0.91 | 56.59 ± 0.91 | 68.94 ± 0.20 | 68.94 ± 0.20 |
| STAD-vMF | 84.32 ± 2.03 | 81.49 ± 4.23 | 56.15 ± 0.98 | 58.02 ± 0.77 | 68.88 ± 0.29 | 71.22 ± 0.40 |

## E.2  STAD IN COMBINATION WITH BN

Table 8: We explored whether STAD, which adapts the classifier, can be effectively combined with TTA methods like BN adaptation, which targets the feature extractor. The results are mixed. On the covariate shift of Yearbook, combining the two methods improves performance beyond what each achieves individually. However, on other datasets, the combination generally results in decreased performance.

| Model | Yearbook covariate shift | + label shift | EVIS covariate shift | + label shift | FMoW covariate shift | + label shift |
|---|---|---|---|---|---|---|
| Source | 81.30 ± 4.18 | | 56.59 ± 0.92 | | 68.94 ± 0.20 | |
| BN | 84.54 ± 2.10 | 70.47 ± 0.33 | 45.72 ± 2.79 | 14.48 ± 1.02 | 67.60 ± 0.44 | 10.14 ± 0.04 |
| STAD-vMF | 85.50 ± 1.34 | 84.46 ± 1.19 | 56.67 ± 0.82 | 62.08 ± 1.11 | 68.87 ± 0.06 | 86.25 ± 1.18 |
| STAD-vMF + BN | 86.20 ± 1.23 | 69.96 ± 0.39 | 44.23 ± 2.88 | 15.18 ± 1.68 | 66.97 ± 0.46 | 9.26 ± 1.97 |
| STAD-Gauss | 86.22 ± 0.84 | 84.67 ± 1.46 | – | – | – | – |
| STAD-Gauss + BN | 86.56 ± 1.08 | 70.12 ±0.33 | – | – | – | – |

## E.3  DOMAIN ADAPTATION BENCHMARKS

To study the limitations and applicability of our method STAD, we also test adaptation performance on non-gradual shifts. For that, we use the domain adaptation benchmark PACS, which comprises images of 10 classes across four categorical domains (*photo*, *art-painting*, *cartoon* and *sketch*). We use DomainBed (Gulrajani & Lopez-Paz, 2021) to train a ResNet-50 model with BN. We test two settings. In the first setting, we follow Iwasawa & Matsuo (2021); Jang et al. (2023) and train the model on three domains and adapt it on the held-out domain. In the second setting, we follow Gui et al. (2024) and train the model on the *photo* domain and adapt it on the remaining domains sequentially. The second setting is more difficult as the model is exposed to only a single domain during training and needs to leverage several abrupt changes in the distribution over a longer test stream.

Tab. 9 shows adaptation performance for the first setting. Adaptation gains are generally smaller than on corruption datasets, with TENT, RoTTA, and SAR improving most upon the source model by 2–3 percentage points. While STAD achieves a more modest improvement of 1 percentage point, it remains the best-performing method among classifier adaptation approaches.

Table 9: Accuracy on **domain adaptation benchmarks** under covariate shift and uniform label distribution. The source model is trained on three domains and tested on the remaining one, rotating the test domain for averaging.

| Method | PACS |
|---|---|
| Source model | $82.99 \pm 8.87$ |
| *adapt feature extractor* | |
| BN | $82.85 \pm 9.57$ |
| TENT | $\mathbf{85.30} \pm 7.33$ |
| CoTTA | $83.59 \pm 8.46$ |
| SHOT | $83.30 \pm 9.01$ |
| SAR | $85.03 \pm 7.71$ |
| RoTTA | $\underline{85.11} \pm 7.73$ |
| *adapt classifier* | |
| LAME | $83.31 \pm 8.90$ |
| T3A | $83.68 \pm 9.14$ |
| **STAD-vMF** (ours) | $83.91 \pm 8.58$ |

Results for the second setting are shown in Tab. 10. Consistent with Gui et al. (2024), we observe a decreasing performance across all TTA methods over the course of adaptation, highlighting the challenge posed by multiple non-gradual domain shifts. Nevertheless, all TTA methods, except LAME, improve upon the source model by over 10 percentage points on average. Despite the highly non-gradual nature of this test setting, STAD-vMF performs comparably to the baselines, achieving the third-best performance overall. These findings strengthens the results in Sec. 5.2, which suggest that STAD is applicable beyond gradual, temporal distribution shifts similar as other TTA methods.

Table 10: Accuracy on **domain adaptation benchmarks** under covariate shift and uniform label distribution. The source model is trained on the photo domain and TTA methods adapt to the remaining domains sequentially. Results show average over three random training seeds. N/A indicates that adaptation is not applied to the source domain.

| Method | P | Domain $\rightarrow$ A | $\rightarrow$ C | $\rightarrow$ S | Mean |
|---|---|---|---|---|---|
| Source | $99.34 \pm 0.57$ | $63.10 \pm 1.55$ | $38.37 \pm 4.99$ | $41.51 \pm 2.99$ | $47.66 \pm 1.29$ |
| *adapt feature extractor* | | | | | |
| BN | N/A | $68.03 \pm 1.98$ | $61.15 \pm 0.34$ | $49.64 \pm 0.28$ | $59.61 \pm 0.52$ |
| TENT | N/A | $68.05 \pm 2.11$ | $61.53 \pm 0.52$ | $51.33 \pm 1.35$ | $60.30 \pm 0.28$ |
| CoTTA | N/A | $63.82 \pm 3.23$ | $59.36 \pm 1.35$ | $56.74 \pm 2.70$ | $59.97 \pm 2.17$ |
| SHOT | N/A | $67.91 \pm 2.04$ | $\mathbf{63.17} \pm 0.86$ | $\mathbf{57.90} \pm 1.22$ | $\mathbf{62.99} \pm 0.58$ |
| SAR | N/A | $68.34 \pm 1.81$ | $61.49 \pm 0.36$ | $52.06 \pm 1.10$ | $60.63 \pm 0.67$ |
| RoTTA | N/A | $\mathbf{68.73} \pm 1.13$ | $58.52 \pm 1.32$ | $52.43 \pm 1.19$ | $59.89 \pm 0.19$ |
| *adapt classifier* | | | | | |
| LAME | N/A | $62.73 \pm 1.72$ | $37.80 \pm 5.09$ | $41.01 \pm 2.90$ | $47.18 \pm 1.26$ |
| T3A | N/A | $68.28 \pm 1.55$ | $\underline{62.05} \pm 0.67$ | $\underline{54.80} \pm 0.97$ | $\underline{61.71} \pm 0.40$ |
| **STAD-vMF** (ours) | N/A | $\underline{68.59} \pm 2.51$ | $61.65 \pm 0.39$ | $52.16 \pm 0.49$ | $60.80 \pm 0.72$ |

### E.4 COMPARISON TO SUPERVISED ORACLE

We investigate how far STAD, which operates unsupervised and does not require labels, can close the gap to a supervised approach that makes use of labels. Obviously, a supervised approach is superior, as it can directly learn the function mapping input samples to target labels. In contrast, TTA methods rely solely on signals from the input and therefore have strictly less information available. To evaluate how far this gap can be bridged, we continuously fine-tune the source model at each timestep using a small portion of labeled samples. For this, we use another held-out set from the Wild-Time pipeline, which is 10% the size of the adaptation test stream. At each timestep, we fine-tune the model for one epoch on this split and then evaluate the performance on the regular test set.

We find that the supervised model achieves an average accuracy of 90.67% over the entire test stream. Comparing this to the source model at 81.30%, STAD-Gauss at 86.22%, and STAD-vMF at

85.50% (see Tab. 2), we observe that STAD can partially close the gap between the unadapted source model and the fine-tuned model. Fig. 7 further reveals that STAD-vMF is on par with the fine-tuned classifier at certain time steps (1986, 1995, 1997). Additionally, we observe that the severity of the shift hampers the supervised model's ability to regain in-distribution accuracy. For instance, in the 1970s and 1980s, the performance of the supervised model is 20 points lower than its in-distribution accuracy (nearly 100%).

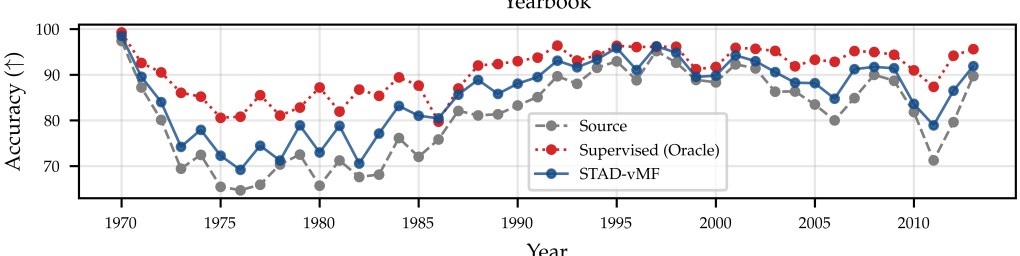

Figure 7: Accuracy over time for the temporal distribution shift on Yearbook averaged over three random training seeds.

## E.5 Cluster Visualization

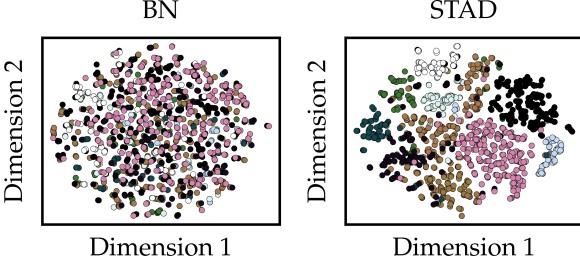

Figure 8: t-SNE visualization of the representation space of FMoW-Time (year 2013) under joint covariate and label shift: We visualize the cluster structure in representation space. Colors indicate ground truth class labels for the 10 most common classes. Adapting with BN destroys the cluster structure, resulting in inseparable clusters. In contrast, STAD operates on linearly separable representations.

## E.6 Runtime

Table 11: Relative runtime per batch compared to the source model: BN and T3A are the fastest for adaptation, while STAD takes significantly less time than CoTTA, which updates all model parameters. Note that baselines like TENT and CoTTA benefit from highly optimized backpropagation code, whereas our codebase has not been optimized for speed. Improving the efficiency of our code would make STAD likely faster.

| Methods | Yearbook | FMoW-Time |
|---|---|---|
| Source Model | 1.0 | 1.0 |
| Batch Norm (BN) | 1.0 | 1.1 |
| TENT | 1.4 | 6.4 |
| CoTTA | 17.1 | 200 |
| SHOT | 1.3 | 6.3 |
| SAR | 1.5 | 7.1 |
| T3A | 1.1 | 1.8 |
| LAME | 1.2 | 2.9 |
| STAD-vMF | 2.5 | 30.8 |
| STAD-Gauss | 3.3 | - |

## E.7 Sensitivity to Hyperparameters

In this section, we conduct a sensitivity analysis of the hyperparameters involved in STAD. We analyze sensitivity on two datasets: the temporal shift dataset Yearbook and the commonly used corruption benchmark CIFAR-10-C. All experiments are conducted under a uniform label distribution. When testing the sensitivity to a specific hyperparameter, all other hyperparameters are fixed at their default values (see App. D). Results show the average over three random training seeds for Yearbook and the average over 15 corruption types for CIFAR-10-C.

**Sensitivity to sliding window size** $s$    The window size determines the number of past time steps considered by the dynamic model. A small $s$ limits the influence of past prototypes, whereas a larger $s$ extends the considered history, giving more weight to past prototypes. However, large values of $s$ come at the cost of increased computational burden, as runtime scales linearly with window size. Tab. 12 suggests that increasing the window size could improve adaptation performance.

Table 12: Accuracy of STAD for different values of $s$

| $s$ | 3 | 5 | 7 |
|---|---|---|---|
| Yearbook | $85.4975 \pm 1.34$ | $85.5022 \pm 1.30$ | $85.5029 \pm 1.31$ |
| CIFAR-10-C | $76.9683 \pm 11.25$ | $76.9735 \pm 11.25$ | $76.9823 \pm 11.24$ |

**Sensitivity to $\kappa^{trans}$** The transition concentration parameter $\kappa^{trans}$ regulates the transition noise and determines how far cluster prototypes move between different time steps. A high concentration value $\kappa^{trans}$ implies little movement of class prototypes, whereas low $\kappa^{trans}$ allows prototypes to move more. This parameter thus acts as a regularization factor between a more static and a more dynamic model. Tab. 13 displays the results. Performance changes only marginally for different values of the concentration parameter.

Table 13: Accuracy of STAD-vMF for different values of $\kappa^{\text{trans}}$

| $\kappa^{\text{trans}}$ | 50 | 100 | 500 | 1000 | 5000 |
|---|---|---|---|---|---|
| Yearbook | $85.5034 \pm 1.3031$ | $85.5034 \pm 1.3031$ | $85.5034 \pm 1.3031$ | $85.5034 \pm 1.3031$ | $85.4980 \pm 1.3099$ |
| CIFAR-10-C | $76.9684 \pm 11.2540$ | $76.9683 \pm 11.2543$ | $76.9685 \pm 11.2538$ | $76.9685 \pm 11.2538$ | $76.9688 \pm 11.2552$ |

**Sensitivity to $\kappa^{ems}$** The emission concentration parameter $\kappa^{ems}$ regulates the emission noise and determines the spread of clusters. A high concentration value $\kappa^{ems}$ implies small, compact clusters, while low $\kappa^{ems}$ allows for widespread clusters. Results are shown in Tab. 14.

Table 14: Accuracy of STAD-vMF for different values of $\kappa^{\text{ems}}$

| $\kappa^{\text{ems}}$ | 50 | 100 | 500 | 1000 | 5000 |
|---|---|---|---|---|---|
| Yearbook | $85.5022 \pm 1.3036$ | $85.5034 \pm 1.3031$ | $85.5043 \pm 1.3019$ | $85.5058 \pm 1.3001$ | $85.5058 \pm 1.3001$ |
| CIFAR-10-C | $76.9679 \pm 11.2543$ | $76.9683 \pm 11.2543$ | $76.9689 \pm 11.2529$ | $76.9700 \pm 11.2514$ | $76.9700 \pm 11.2513$ |

