# OpenReview forum: "Temporal Test-Time Adaptation with State-Space Models"
_ICLR.cc/2025/Conference — Submitted to ICLR 2025_

### Official Review · Reviewer_2f87 · 2024-10-17

**Soundness:** 3
**Presentation:** 3
**Contribution:** 2
**Rating:** 6
**Confidence:** 4

**Summary:**

This paper proposes a method STAD designed to adapt a model’s classification head during inference to handle temporal distribution shifts, which are underexplored. The paper focuses on test-time adaptation, proposing to track the evolution of hidden features using SSMs. STAD dynamically updates class prototypes to reflect the evolving data distribution. Extensive experiments demonstrate that STAD performs well in temporal distribution shifts, label shifts, and small batch size settings. The paper evaluates its approach on several real-world datasets, showing robustness and performance gains compared to baseline methods.

**Strengths:**

1. The paper is generally well-written with clear explanations of both the problem and the proposed solution.
2. The authors provide a strong theoretical foundation for the use of probabilistic SSMs to address temporal test-time adaptation, with detailed mathematical formulations of the dynamics model.
3. The empirical analysis is comprehensive, with evaluations on multiple datasets, including both real-world temporal shifts and synthetic corruption shifts.

**Weaknesses:**

1. The paper does not clearly describe the sensitivity of STAD to hyperparameters which could impact the method’s robustness.
2. While STAD adapts the last layer to changing distributions, the reliance on class prototypes could be problematic when distributions are overfitted to the target domains and cause catastrophic forgetting.
3. The gradual distribution shift assumption is pretty strong especially during test time, which undermines the significance of the method. For example, ATTA [1] also targets distribution shifts during test time, which also includes some similar distribution shifts constrains. Given these constrains, they reason that a few labeled data is necessary to make the setting reasonable in real world scenarios. As what I believe, the gradual distribution shift assumption is a major limitation of this work. While it may be hard to make this method work perfectly on non-trivial domain shift datasets such as PACS and VLCS, there are two important experiments to do. (1)  Building boundaries on the ability limitations and applicability of this work on non-trivial domain shift datasets. (2) Validating that it is **not necessary** to use labels in the gradual shift setting by showing that STAD's performance is similar to the methods using labels (as Oracle), such as active online learning and ATTA.

[1] Gui, Shurui, Xiner Li, and Shuiwang Ji. "Active Test-Time Adaptation: Theoretical Analyses and An Algorithm." In The Twelfth International Conference on Learning Representations.

**Questions:**

How can we quantify the gradual distribution shifts?

---

> ### Author Response · Authors · 2024-11-21
> **Rebuttal**
>
> We appreciate your feedback and the effort you put into reviewing our work. We address each comment below.
>
> > The paper does not clearly describe the sensitivity of STAD to hyperparameters which could impact the method’s robustness.
>
> We analyzed the sensitivity of STAD’s hyperparameters (concentration parameters $\kappa^{trans}, \kappa^{ems}$ and window size $s$) in Appendix D5. In the revised version of the paper, we have added additional explanations and provided sensitivity analyses of hyperparameters on CIFAR-10-C as well. If this does not address your concern, please let us know what you feel is missing.
>
> > While STAD adapts the last layer to changing distributions, the reliance on class prototypes could be problematic when distributions are overfitted to the target domains and cause catastrophic forgetting.
>
> In our response here, we assume you are concerned about adapted prototypes becoming increasingly different from the source prototypes over the course of adaptation. Let us know if we have misunderstood.
>
> Keeping the balance between agile adaptation and catastrophic forgetting is a tradeoff in every TTA method. In STAD, catastrophic forgetting is addressed through the following principles:
> 1. **Choice of adaptable layers**: Keeping the feature extractor fixed and adapting only the last layer serves as regularization and helps prevent catastrophic forgetting. Experimentally, we show this in Table 1, which reveals that feature-extractor adaptation methods collapse significantly while classifier-adaptation methods yield stable performance across all shift settings.
> 2. **Use of filtering theory:** The tradeoff between incorporating new information and retaining existing knowledge has been long studied in the filtering literature, with the Kalman filter being provably the optimal solution for the linear filtering problem [1]. By building on state-space models such as the Kalman filter, STAD is given a built-in mechanism to control the impact of new batches on the adapted weights in a principled manner.
> 3. **EM-algorithm initialization:** In STAD, the source weights act as a prior at each time step (Algorithm 1, line 2). This means the source knowledge is constantly reinforced as a prior (this prevents forgetting), while the learned dynamics of the test data enter via the transition model (Eq 25, 28 for STAD-vMF) (this ensures adaptability).
>
> [1] Rudolph Emil Kalman. A new approach to linear filtering and prediction problems. 1960

---

> > ### Author Response · Authors · 2024-11-21
> > **Rebuttal (continued)**
> >
> > > The gradual distribution shift assumption is pretty strong, especially during test time, which undermines the significance of the method. For example, ATTA [1] also targets distribution shifts during test time, which also includes some similar distribution shift constraints. Given these constraints, they reason that a few labeled data points are necessary to make the setting reasonable in real-world scenarios. As what I believe, the gradual distribution shift assumption is a major limitation of this work. While it may be hard to make this method work perfectly on non-trivial domain shift datasets such as PACS and VLCS, there are two important experiments to do:
> > > (1) Building boundaries on the limitations and applicability of this work on non-trivial domain shift datasets.
> > > (2) Validating that it is **not necessary** to use labels in the gradual shift setting by showing that STAD's performance is similar to the methods using labels (as Oracle), such as active online learning and ATTA.
> >
> > Thank you for the precise suggestions. We agree with your view that labels are required for adapting to strong shifts. Blindly adapting to such shifts can lead to model collapse and can cause severe consequences in safety-critical deployment scenarios. This is why we monitor adaptation with dispersion (see Section 5.3, Figure 6 (right)), which can alert us when adaptation fails and supervised training is required.
> >
> > It seems your critique is directed towards the line of research on TTA in general and not the particularities of our paper, which appears unfair. In general, we agree there is no free lunch. If labeled data is not available, shifts must be relatively bounded to significantly improve upon the source model. We are following a long line of work in TTA that studies this scenario: **according to Xiao & Snoek [1], there have been over 200 papers on TTA since 2020**.
> >
> > **Regarding (1):** We highlight that the temporal distribution shift datasets we use in our experiments are non-trivial. The Wild-Time benchmark (NeurIPS 2022), which we test on, has been widely used (>80 citations) in related fields such as domain generalization and domain adaptation. To answer your question on the adaptability of STAD to domain adaptation benchmarks, we ran experiments on PACS and reported results in Appendix E.3 of the revised paper. We found that STAD improves less significantly upon the source model on this dataset (1 ppt in accuracy), yet performs similarly to some of the TTA baselines.
> >
> > **Regarding (2):** Testing whether having labels yields better performance than not having labels seems not a relevant hypothesis for this work. Clearly, having labels provides a stronger learning signal. However, our work studies unsupervised adaptation where having labels is not an option.
> >
> > > How can we quantify the gradual distribution shifts?
> >
> > Past theoretical work has quantified gradual distribution using different divergence functions. However, our work does not aim to provide theoretical insights but addresses the practical problem of real-world shifts over time. To measure when the gradual shift assumption is violated and adaptation is not encouraged, we use the dispersion metric (Section 5.3) to alert failures of adaptation.
> >
> > We hope we have clarified your concerns. If that’s the case, we would appreciate it if you would consider revising your score. If there are any remaining unaddressed concerns, we are happy to provide further clarifications.
> >
> > [1] Xiao, Zehao, and Cees GM Snoek. "Beyond Model Adaptation at Test Time: A Survey." arXiv preprint arXiv:2411.03687 (2024).

---

> ### Comment · Reviewer_2f87 · 2024-11-22
>
> Thank you for the detailed rebuttals and extensive efforts.
>
> I am satisfied with the response to the first concern, however,
>
> 1. The catastrophic forgetting and overfitting issues are not addressed well.
> 2. The authors mentioned that "there have been over 200 papers on TTA since 2020." That is true, but today's TTA development focuses more and more on catastrophic forgetting and model collapse issues since they are the obstacles to the use of TTA techniques in the real world. Therefore, it is totally fair to question your methods and assumptions on these issues. The gradual distribution shift assumption is also questioned by Reviewer HZGZ.
>
> Therefore, I suggested several ways to make your experiments and method robust.
>
> (1) Evaluating the method on large shift datasets like PACS, but the authors did not benchmark the method in a comparable continuous setting. For example, training on P and testing on A, C, and S sequentially.
> (2) Evaluating the method with a more restrictive setting with supervised signals is a great way to show that "our method does not need these signals but performs similarly or closed". This is an important step to convince people that **this setting makes sense**. Although this setting/assumption was questioned by more than one reviewer, the authors refused to conduct such important experiments.
> (3) Quantifying the distribution shifts of the datasets you used in experiments is also a significant way to show readers that **the evaluated distribution shifts are not trivial**. Nevertheless, instead of trying to convince readers that the gradual distribution shifts are non-trivial compared to other shifts by quantifying them, the authors claimed that they are not aiming to provide theoretical insights, which is not related to what we care about.
>
> I appreciate and acknowledge the efforts the authors made, but the current rebuttal is not convincing. In addition, please highlight the changes in your revised version for reviewers' convenience.

---

> ### Author Response · Authors · 2024-11-25
>
> Thank you for the continuous engagement and detailed comments, we appreciate it a lot. We have updated our revised version, with changes to the original submission highlighted in red.
>
> > Evaluating the method on large shift datasets like PACS, but the authors did not benchmark the method in a comparable continuous setting. For example, training on P and testing on A, C, and S sequentially.
> >
>
> Thanks for clarifying the evaluation setting you are interested in. We have added it to the revised version. In Appendix E.3, we now test the classic DomainBed [1] setting also used by [2,3] (train on three domains, adapt on remaining one) and the sequential setting used by the ATTA paper [4] (train on P, sequentially adapt on remaining domains). We find that despite the non-gradual nature of the sequential setting, STAD performs competitively to other TTA methods. This is consistent with results in Section 5.2., where we find that STAD can be applied beyond the temporal shift setting, which is the main focus of our work.
>
> > Evaluating the method with a more restrictive setting with supervised signals is a great way to show that "our method does not need these signals but performs similarly or closed".
> >
>
> We have added a comparison with a supervised oracle model on Yearbook to Appendix E.3. As oracle, we report test performance of the source model continuously fine-tune each time step with 10% of additional labeled data. Results are visualised in Figure 7. We find that our method (accuracy 86 %) can close approximately half of the gap between the unadapted model (accuracy 81 %) and the supervised model (accuracy 91%). Just to clarify, **we do not claim our method (or any TTA method) should perform similar to the supervised method since it relies on unlabelled data and therefore weaker learning signals.** Instead, as we can see it can help to improve source model performance when labels are not available (+5% here).
>
> We thank the reviewer for this suggestion as we think it nicely highlights the capabilities and limits of TTA in general.
>
> > Quantifying the distribution shifts of the datasets you used in experiments is also a significant way to show readers that **the evaluated distribution shifts are not trivial**.
> >
>
> Thanks for clarifying why you wish to quantify the distribution shift. It was not clear to us previously that you aim for a measure to assess if the datasets we test on are non-trivial. The severeness of the distribution shift can be quantified in terms of accuracy drop of the source model. We provide these results in Figure 4. Figure 4 (as well as newly added Figure 7) show the severe drop in accuracy of the source model when exposed to temporal distribution shifts. Concretely, accuracy drops by up to 30 points on Yearbook (98 % to as low as 67%), 8 points on EVIS (62% to as low as 54%) and 33 points on FMoW (81% to as low as 48%). We believe these substantial drops in the performance of source models highlight the fact that the **gradual distribution shifts considered in our work are non-trivial**.
>
> > The catastrophic forgetting and overfitting issues are not addressed well
> >
>
> In our past response we have listed mechanisms of STAD that aim to prevent catastrophic forgetting. To experimentally demonstrate the effectiveness of these mechanisms, we apply our method on the source domain after adapting to the target domain. The table below shows accuracy on Yearbook. We find no signs of catastrophic forgetting as the performance on the source domain is not diminished by STAD. In fact, STAD has a small positive impact on the source domain performance.
>
> |  | target domains (1970-2014) | → source domains (1930-1969) |
> | --- | --- | --- |
> | Source | 81.30 | 99.41 |
> | STAD-vMF | 85.50 | 99.58 |
>
> Please let us know if this resolves your concerns regarding catastrophic forgetting. Otherwise, we would appreciate if you could clarify the question.
>
> Let us know if there are any further unaddressed concerns. We thank you again for your continued engagement.
>
> ---
>
> [1] Gulrajani, Ishaan, and David Lopez-Paz. "In search of lost domain generalization." ICLR 2021.
>
> [2] Iwasawa, Yusuke, and Yutaka Matsuo. "Test-time classifier adjustment module for model-agnostic domain generalization." NeurIPS 2021.
>
> [3] Jang, Minguk, Sae-Young Chung, and Hye Won Chung. "Test-time adaptation via self-training with nearest neighbor information." ICLR 2023.
>
> [4] Gui, Shurui, Xiner Li, and Shuiwang Ji. "Active test-time adaptation: Theoretical analyses and an algorithm." ICLR 2024.

---

> > ### Comment · Reviewer_2f87 · 2024-11-26
> >
> > Thank you for the authors' extensive efforts and revision. The updated version is much clearer regarding the performance limitations and the method's applicability. I have raised my score from 5 to 6.

---

> > > ### Author Response · Authors · 2024-11-27
> > >
> > > Thank you for increasing your score! We are encouraged that your concerns appear to have been resolved by the supporting experiments included in the Appendix. To summarize above discussion, we are particularly glad that we could provide convincing arguments that
> > >
> > > - our label-free method is naturally upper-bounded by a supervised oracle. This is a **limitation that also applies to every other TTA method**.
> > > - **real-world shifts that occur gradually over time are non-trivial** (see Figure 4). This justifies their investigation, which is precisely the aim of our work.
> > > - our method is **applicable beyond gradual shifts** and performs comparably to baselines in non-gradual settings (see Table 3, 9 and 10)
> > >
> > > We are grateful for your engagement in this discussion, it surely helped us highlight important points of our work.

---

### Official Review · Reviewer_HZGZ · 2024-11-01

**Soundness:** 3
**Presentation:** 3
**Contribution:** 2
**Rating:** 3
**Confidence:** 4

**Summary:**

This paper addresses the test-time adaptation problem under gradual distribution shifts. The proposed method models the gradual distribution shift in the representation space using linear state-space models, implemented through the von Mises-Fisher distribution. Experimental results on multiple real-world datasets demonstrate the effectiveness of the proposed algorithm.

**Strengths:**

This work conducts experiments on several real-world datasets, enhancing the practical applicability of TTA algorithms in real-world settings.

By modeling gradual distribution shifts using linear state-space models, this work provides fresh insights in the field of TTA.

**Weaknesses:**

The paper lacks discussion and empirical comparisons with related work, particularly in the field of gradual unsupervised domain adaptation, which addresses very similar problems (e.g., [1, 2]). As a result, the contribution of this paper may be overstated.

The gradual shift assumption represents a relatively simple form of distribution shift in dynamic environments, as its total variation is small. This simplicity limits the contribution of the work.

The proposed method appears to combine existing approaches, which limits its technical novelty.

References:

[1] Kumar, Ananya, Tengyu Ma, and Percy Liang. "Understanding self-training for gradual domain adaptation." International Conference on Machine Learning. PMLR, 2020.
[2] Wang, Haoxiang, Bo Li, and Han Zhao. "Understanding gradual domain adaptation: Improved analysis, optimal path, and beyond." International Conference on Machine Learning. PMLR, 2022.

**Questions:**

How does the proposed method compare to self-training algorithms, e.g., Ref [1], on real-world tasks?

---

> ### Author Response · Authors · 2024-11-21
> **Rebuttal**
>
> We thank you for your time in reviewing our work.
>
> > The paper lacks discussion and empirical comparisons with related work, particularly in the field of gradual unsupervised domain adaptation, which addresses very similar problems (e.g., [1, 2]). As a result, the contribution of this paper may be overstated.
> >
>
> Thank you for the references on gradual domain adaptation (GDA).
>
> We do see the similarities between GDA and the setting we study. However, we wish to highlight that GDA requires access to a labeled source dataset. In real-world applications, this is often unavailable, motivating the well-established test-time adaptation setting that our work focuses on. Methods of GDA are thus not applicable to our experiments.
>
> We have incorporated a discussion on domain generalization (DG) and domain adaptation into the related work (Section 4). There, we discuss links to related settings (temporal DG, GDA), their similarities (e.g., gradual shift), and differences (e.g., requiring access to source data) compared to our setting.
>
> > The gradual shift assumption represents a relatively simple form of distribution shift in dynamic environments, as its total variation is small. This simplicity limits the contribution of the work.
> >
>
> We respectfully disagree that this assumption is a limitation: unsupervised adaptation is all but impossible without the shifts being relatively modest. Yet many real-world applications exhibit such bounded shifts, as demonstrated by our method’s success on Wild-Time benchmarks.
>
> - **“Total variation is small”**: Small variation keeps the problem tractable in an unsupervised setting. As you pointed out, this tractability has been exploited by the gradual domain adaptation literature [1, 2].
> - **“Simple form of distribution shift in dynamic environments”**: Gradual shifts are pervasive, particularly in dynamic environments. These environments are characterized by the continuous passage of time, making gradual changes very plausible. Well-studied examples include weather changes in autonomous driving [3], the evolution of research fields [4], and opinion drift on social media [5].  In contrast, sudden shifts, such as domain shifts or subpopulations shifts, are predominant when the environment is static. Static environments are described by categorical factors (e.g., different hospitals) rather than a continuous domain axis, such as time.
> - **“This simplicity limits the contribution of the work”**: Past work has mostly provided considerable adaptation gains on image corruptions. Image corruptions increase the degree of noise while the underlying signal remains static. In the gradual shifts that we study, the signal changes and not the noise. This structural change can arguably be considered as the more difficult setting as opposed to denoising.
>
> > The proposed method appears to combine existing approaches, which limits its technical novelty.
> >
>
> We respectfully disagree. Please note that the reviewer guidelines explicitly state that a combination of existing ideas can provide an original and significant contribution:
>
> *We encourage reviewers to be broad in their definitions of originality and significance. For example, originality may arise from a new definition or problem formulation, **creative combinations of existing ideas**, application to a new domain, or removing limitations from prior results.*
>
> To the best of our knowledge, our work is the first to address test-time adaptation using state-space models. If there is prior work, we would appreciate it if you could provide a reference.
>
> > How does the proposed method compare to self-training algorithms, e.g., Ref [1], on real-world tasks?
> >
>
> Please note that the referenced work [1] is not applicable in our setting, as it assumes simultaneous access to labeled source and unlabeled target data. Specifically, [1] trains the model using hinge or ramp loss (see Section 3.1), which requires modifying the training procedure and access to a labeled source dataset. The test-time adaptation (TTA) setting we study uses off-the-shelf pre-trained models and assumes access only to an unlabeled target dataset.
>
> Additionally, the baselines we evaluate already incorporate forms of self-training. For example, CoTTA employs a student-teacher framework, and TENT uses entropy minimization. Results for these methods are shown in Tables 1, 2, and 4, as well as Figures 4 and 5.
>
> If we have clarified your concerns, we would appreciate your consideration in raising the score.
>
> [1] Kumar, Ananya, et al. "Understanding self-training for gradual domain adaptation." ICML 2020.
>
> [2] Wang, Haoxiang, et al. "Understanding gradual domain adaptation." ICML 2022.
>
> [3] Gong, Taesik, et al. "Robust continual test-time adaptation against temporal correlation." NeurIPS 2022.
>
> [4] Yao, Huaxiu, et al. "Wild-Time: A benchmark of in-the-wild distribution shift over time." NeurIPS 2022.
>
> [5] Cai, Zekun, et al. "Continuous Temporal Domain Generalization." NeurIPS 2024.

---

> > ### Author Response · Authors · 2024-11-25
> >
> > Dear Reviewer HZGZ,
> >
> > Thank you again for your comments. We would greatly appreciate it if you could share any remaining thoughts during the last 2 days of the discussion period to indicate whether our rebuttal has addressed your concerns.
> >
> > We remain available to respond to any additional questions.
> >
> > Thanks,
> >
> > Authors

---

> > > ### Author Response · Authors · 2024-11-30
> > >
> > > Dear Reviewer HZGZ,
> > >
> > > Thanks again for your efforts in reviewing our paper.
> > >
> > > We would appreciate if you could let us know your thoughts on our rebuttal.
> > >
> > > Thanks,
> > >
> > > Authors

---

### Official Review · Reviewer_PoMj · 2024-11-03

**Soundness:** 1
**Presentation:** 2
**Contribution:** 2
**Rating:** 3
**Confidence:** 4

**Summary:**

This paper introduces a novel Test-Time Adaptation (TTA) method, STAD, which leverages a state-space model to learn time-evolving feature distributions. It also proposes a new TTA setting that is more reflective of real-world scenarios.

**Strengths:**

1. The authors propose a realistic TTA setting and conduct extensive experiments across multiple datasets to validate the method's effectiveness.
2. A new TTA method, STAD, is introduced, combining state-space models with a solid theoretical foundation.

**Weaknesses:**

1. The method modifies and updates only the linear classifier, raising concerns about effectively handling covariate shifts. As shown in Table 4, adapting only the classifier on CIFAR-10C performs significantly worse than adapting the feature extractor.

2. Experimentally, while a realistic TTA setting is proposed, comparisons with other similar settings and related methods are lacking. For instance, the following related works are not adequately compared:

   1. UniTTA: Unified Benchmark and Versatile Framework Towards Realistic Test-Time Adaptation. arXiv 2024.
   2. Towards real-world test-time adaptation: Tri-net self-training with balanced normalization. AAAI 2024.
   3. Robust test-time adaptation in dynamic scenarios. CVPR 2023.
   4. Universal Test-Time Adaptation Through Weight Ensembling, Diversity Weighting, and Prior Correction. WACV 2024.

   Additionally, only a small portion of common TTA datasets (CIFAR-C, ImageNet-C) are addressed, with limited focus on CIFAR-10C.

**Questions:**

See the weaknesses section.

---

> ### Author Response · Authors · 2024-11-21
> **Rebuttal**
>
> We thank you for the comments and references. We address each concern below.
>
> > Soundness: 1: poor
> >
>
> We noticed that you rated the soundness of our paper as “poor” while simultaneously acknowledging the “solid theoretical foundation” of our method and our “extensive experiments.” This appears to be self-contradictory. Could you please explain the reasoning behind this rating?
>
> > The method modifies and updates only the linear classifier, raising concerns about effectively handling covariate shifts. As shown in Table 4, adapting only the classifier on CIFAR-10C performs significantly worse than adapting the feature extractor.
> >
> >
> > Additionally, only a small portion of common TTA datasets (CIFAR-C, ImageNet-C) are addressed, with limited focus on CIFAR-10C.
> >
>
> We respectfully disagree. First, a key goal of this work is to move beyond commonly used image corruption benchmarks, such as CIFAR-10/100-C and ImageNet-C. The limited diversity of benchmarks in TTA has been noted in prior work [5]. Additionally, our emphasis on real-world distribution shifts has been positively acknowledged by all reviewers as, for instance, "enhancing the practical applicability of TTA algorithms" (reviewer HZGZ) and "underexplored" (reviewer 2f87).
>
> Second, while image corruption datasets have predominated in TTA research, feature-extractor adaptation methods have excelled in this context because they address shifts in earlier layers [6]. There are pros and cons to both feature-extractor and classifier adaptation methods, and our paper provides nuanced insights into these trade-offs. On image corruptions (Table 4), feature-extractor adaptation methods perform better overall. (Notably, our method is the best-performing classifier adaptation method, outperforming established methods like T3A and LAME.)
>
> However, as seen in Table 1, feature-extractor adaptation methods fail significantly on most real-world temporal distribution shifts, the focus of our work. This highlights the effectiveness of classifier adaptation methods in addressing temporal shifts.
>
> > Experimentally, while a realistic TTA setting is proposed, comparisons with other similar settings and related methods are lacking. For instance, the following related works are not adequately compared:
> >
> > 1. UniTTA: Unified Benchmark and Versatile Framework Towards Realistic Test-Time Adaptation. arXiv 2024.
> > 2. Towards real-world test-time adaptation: Tri-net self-training with balanced normalization. AAAI 2024.
> > 3. Robust test-time adaptation in dynamic scenarios. CVPR 2023.
> > 4. Universal Test-Time Adaptation Through Weight Ensembling, Diversity Weighting, and Prior Correction. WACV 2024.
>
> Thank you for these references.
>
> **On differences in settings**
>
> We wish to emphasize that while the referenced works—like ours—advocate for more realistic evaluations of TTA methods, they do so in a fundamentally different way. Past work focuses primarily on **diversity in the ordering of samples in the test stream**, relying on common datasets (primarily image corruptions). In contrast, our work emphasizes **dataset diversity**, targeting real-world distribution shifts.
>
> For clarity, we have dedicated Appendix C to a detailed discussion of differences between our work and prior approaches aiming for realistic TTA evaluation.
>
> **On associated TTA methods**
>
> - **ROID [4]:** The ROID setting introduces mixed domains per batch, which is incompatible with our work. Our domain index is temporal, and ordering is determined by timestamps. Mixing temporal domains would destroy this inherent ordering, undermining the core motivation of our work.
> - **UniTTA [1]:** This concurrent work is also designed for the mixed domain setting and not directly comparable to our setting.
> - **TRIBE [2]:** This work addresses global class imbalance, which is outside the scope of our research.
> - **RoTTA [3]:** We discuss this work in our related work section. RoTTA introduces temporal correlations between samples with continuously changing distributions. We added RoTTA to our baselines, searching learning rates {1e-3, 1e-4, 1e-5}, and reported the best results. Table 1 shows that while RoTTA mitigates extreme collapse in the label shift setting (as claimed in their paper), it **fails to improve upon the source model across all temporal shift datasets.** This aligns with our prior findings.
>
> If you think we have adequately addressed your concerns, we would appreciate you considering raising your score.
>
> [5] Zhao, Hao, et al. "On pitfalls of test-time adaptation." ICML 2022.
>
> [6] Lee, Yoonho, et al. "Surgical fine-tuning improves adaptation to distribution shifts." ICLR 2023.

---

> > ### Author Response · Authors · 2024-11-25
> >
> > Dear Reviewer PoMj,
> >
> > Thanks again for your efforts. We appreciate if you could use the last 2 days of the discussion period to let us know if the concerns you raised have been addressed by our rebuttal.
> >
> > We remain available for potential follow-up questions.
> >
> > Thank you,
> >
> > Authors

---

> ### Author Response · Authors · 2024-11-30
>
> Dear Reviewer PoMj,
>
> Thanks again for your efforts in reviewing our paper.
>
> We would appreciate if you could let us know your thoughts on our rebuttal.
>
> Thanks,
>
> Authors

---

### Official Review · Reviewer_QpBz · 2024-11-03

**Soundness:** 3
**Presentation:** 3
**Contribution:** 3
**Rating:** 6
**Confidence:** 3

**Summary:**

The paper studied test-time adaptation, a setting where model parameters are updated based on incoming test features. It proposes STAD, a method to track gradual distribution shifts which only updates the last layer in a given network, based on the EM algorithm. The authors report results on several tasks.

**Strengths:**

The paper is well-written and gives a principled justification for the proposed method. It considers real-world shifts and the proposed method seems to outperform the baselines generally.

**Weaknesses:**

The paper does not appear to have any major weaknesses, but I am curious about the importance of the subfield (temporal test-time adaptation). Why is this the right setting to study (as opposed to others for distribution shift) and how does it compare to methods involving periodic retraining and unsupervised domain adaptation?

**Questions:**

LIne 37 - why can't test-time training operate in this paradigm? what is the precise difference between TTA and TTT?
46 - Doesn't FMoW have labels? Why is this dataset suitable for the proposed setting, which claims that one of its benefits is not needing labels?
192 - missing a comma inside the parentheses?

How important is the linear Gaussian transition model assumption? Could your method make use of labels if they were made available?

---

> ### Author Response · Authors · 2024-11-21
> **Rebuttal**
>
> Thank you for the positive assessment of our work. We are encouraged that you identified no major weaknesses and appreciate your recognition of the principal justification of our method as well as our focus on real-world shifts. Below, we respond to each of your questions.
>
> > How does [the setting] compare to methods involving periodic retraining and unsupervised domain adaptation?
> >
> >
> > Line 37 - why can't test-time training operate in this paradigm? What is the precise difference between TTA and TTT?
> >
>
> Test-time adaptation (TTA) has received considerable attention in recent years [1] as a particularly flexible solution for resource-constrained environments. It contrasts with related approaches aimed at generalizing across domains, as it requires only a few test instances and the trained model:
>
> - **Domain Generalization** focuses on improving training algorithms for better out-of-distribution generalization and operates entirely during training, without adaptation during deployment.
> - **Periodic Retraining** maintains model performance by retraining on labeled samples from the new domain. However, it requires labeled samples, which can be costly to obtain, especially in dynamic environments with continuously changing data.
> - **Unsupervised Domain Adaptation (UDA)** aligns target data with training data, requiring access to training data and a large amount of unlabeled target data, making it more suitable for offline settings.
> - **Test-Time Training (TTT)** adapts model parameters based on unlabeled test data, typically using gradient steps on an auxiliary task (e.g., predicting image rotations). However, this auxiliary task is introduced during training, meaning TTT is not agnostic to the training process.
>
> In contrast, TTA operates online, adapting a pre-trained model directly at test time using only a few unlabeled test samples. It does not rely on training data or the original training process, allowing it to work “out of the box” with any pre-trained model. This flexibility has driven significant research in recent years [1] and makes TTA widely applicable. In the revised version of the manuscript we have made these differences more clear (see also global rebuttal response).
>
> > The paper does not appear to have any major weaknesses, but I am curious about the importance of the subfield (temporal test-time adaptation). Why is this the right setting to study (as opposed to others for distribution shift)?
> >
>
> Most prior work on TTA has focused on image corruptions like those in CIFAR-10-C, where distribution shifts are primarily information degradation (e.g., blurring, noise). While these benchmarks have advanced TTA, they assume a fixed underlying signal that simply becomes noisier over time. However, real-world environments are often subject to structural changes rather than just information loss, and such changes are inherently difficult to predict in advance. This makes TTA the ideal paradigm to study temporal distribution shifts, as it allows for online adjustment.
>
> We note that the importance of temporal distribution shifts has been widely recognized in related fields, such as domain generalization [2] and domain adaptation [3]. Our work highlights a gap in TTA research, as our experiments show that many existing TTA methods are ill-equipped to handle this ubiquitous type of shift effectively.
>
> > 46 - Doesn't FMoW have labels? Why is this dataset suitable for the proposed setting, which claims that one of its benefits is not needing labels?
> >
>
> Yes, FMoW has labels. However, the adaptation and model predictions are solely based on unlabeled input images. Test labels are only used to compare model predictions with the ground truth to assess the performance of different TTA methods.
>
> > How important is the linear Gaussian transition model assumption?
> >
>
> Thank you for the question. The linear transition assumption provides a balance between adaptability and regularization. A more flexible, non-linear transition model could risk overfitting, especially since the dynamics are learned entirely unsupervised.
>
> > Could your method make use of labels if they were made available?
> >
>
> This work focuses on unsupervised adaptation and STAD has no mechanism to incorporate labels. However, since TTA is only effective when the underlying signal does not change entirely, it would be interesting to explore in future work when labels are required and when the limits of TTA are reached.
>
>
> [1] Xiao, Zehao, and Cees GM Snoek. "Beyond Model Adaptation at Test Time: A Survey." arXiv preprint arXiv:2411.03687 (2024).
>
> [2] Bai, Guangji, Chen Ling, and Liang Zhao. "Temporal domain generalization with drift-aware dynamic neural networks." ICLR 2023.
>
> [3] Kumar, Ananya et al. "Understanding self-training for gradual domain adaptation." ICML 2020.

---

### Author Response · Authors · 2024-11-21
**Rebuttal**

We thank all reviewers for the time and expertise they have invested in their reviews and their insightful comments and suggestions.

We are pleased to hear that our paper offers “fresh insights in the field of TTA” (HZGZ), has “a strong/solid theoretical foundation” (2f87, PoMj) and provides “principled justifications” (QpBz) for our “novel” method (PoMj). Reviewers appreciated our focus on temporal distribution shifts, which are “underexplored” (2f87), “more reflective of real-world scenarios” (PoMj) and helpful in “enhancing the practical applicability of TTA algorithms in real-world settings” (HZGZ). We are delighted that our “empirical analysis is comprehensive” (2f87) and we could demonstrate “robustness and performance gains compared to baseline models” (2f87, QpBz) via “extensive experiments” (2f87, PoMj). Lastly, our paper is “well-written” (QpBz,2f87), provides “clear explanations of both the problem and the proposed solution” (2f87) with “detailed mathematical formulations” (2f87).
# Clarification of the Setting
The only common point raised in the reviews (reviewers QpBz, HZGZ, 2f87) relates to the link between our precise setting and adjacent settings, particularly the link to gradual unsupervised domain adaptation (reviewer HZGZ) and TTA with supervision (reviewer 2f87).

These two settings have strictly more information at hand than test-time adaptation. Gradual domain adaptation assumes simultaneous access to the source and target data. Active test-time adaptation assumes it is possible to query labels at test time. In real-world settings, these assumptions may easily not be met. Privacy concerns or model access via an API can prevent access to the source. In dynamic environments where instant predictions are required, an intermediate annotating process may not be feasible. These restrictions have motivated the long line of test-time adaptation research (see [1,2] for surveys).

In our study, we aim to replicate the simple yet widespread and practical setting where a model is trained and then deployed to the real world without prior knowledge of the distribution shifts it may encounter. When the shift is too strong, retraining with supervision (e.g., with active TTA) is clearly necessary. However, when the shift is mild, there is an opportunity to keep the model running. This is the setting we address. We believe we achieve this effectively by using the well-established Wild-Time Benchmarks (NeurIPS 2022, >80 citations, used by, e.g., [3]), which were explicitly designed to capture real-world distribution shifts and realistic deployment scenarios.

To bring more clarity in distinguishing our problem setting from adjacent settings, we have expanded the related work section (Section 4) with a discussion of domain generalization and domain adaptation. We have also expanded the setting comparison in Table 1.

# Changes to the Paper
We have made minor changes to the paper to address the reviewers' comments:
- We have **added RoTTA as an additional baseline** (Tables 2, 3, and 4). Results confirm the experimental trends already observed: RoTTA fails to improve on all temporal distribution shifts.
- We have **added a discussion of adjacent fields** domain generalization and unsupervised domain adaptation to the related work (Section 4) and the setting comparison (Table 1).
- We have **added a more detailed discussion of additional TTA work** in Appendix C.
- We have **added another dataset** from domain generalization (Appendix E.3).
- We have **expanded the sensitivity study** in Appendix E.6.

Our paper now includes experiments on **4 types of distribution shifts** (temporal, reproduction, corruption, and style transfer) across **7 datasets** comparing against **8 baselines**. We further evaluate robustness on 12 different batch sizes for 2 label shift settings, in addition to an ablation study and qualitative results. Furthermore, we anchor our paper in past work by linking it to four related fields with additional in-depth discussion in the Appendix. We are positive about our theoretically grounded approach being well evaluated and embedded in past work.

[1] Liang, Jian, Ran He, and Tieniu Tan. "A comprehensive survey on test-time adaptation under distribution shifts." International Journal of Computer Vision (2024): 1-34.

[2] Xiao, Zehao, and Cees GM Snoek. "Beyond Model Adaptation at Test Time: A Survey." arXiv preprint arXiv:2411.03687 (2024).

[3] Zhao, Hao, et al. "On pitfalls of test-time adaptation." ICML 2023. (see https://github.com/LINs-lab/ttab/tree/main)

---

### Meta-Review · Area_Chair_CMJS · 2024-12-22

**Metareview:**

The paper introduces STAD, a method leveraging state-space models for test-time adaptation under gradual distribution shifts. Addressing temporal shifts is of interest for robust learning. The paper provides comprehensive experiments and the results are promising. However, reviewers identified critical issues such as missing comparisons with related work and over-reliance on a simplified shift assumption. I find the amount of missing references to be concerning. There is also a lack of formal mathematical guarantees on the method (e.g. when the method provably works, generalization bounds etc). While I recommend rejection for the current submission, I believe this is a promising contribution and I recommend authors to address these issues. That is, (1) carefully incorporate the related literature, (2) provide stronger justification on when/why the stated assumptions are necessary, and (3) explore whether formal guarantees can be provided on the method e.g. what level of temporal shift it can tolerate or when it provably fails.

**Additional Comments On Reviewer Discussion:**

During the rebuttal, the authors addressed feedback by extending experiments, adding baselines, and clarifying methodological details. However, this did not solve all issues (see the meta-review for details). I was specifically concerned by the amount of missing references which were added later. It is possible that there are still missing comparisons/literature which is extremely challenging to verify during the rebuttal period. A resubmission and proper review process makes sense in that regard. I should also remark that two of the reviewers unfortunately did not follow up with authors.

---

### Decision · Program_Chairs · 2025-01-22

Reject